# iSCAN: Identifying Causal Mechanism Shifts among Nonlinear Additive Noise Models

**Tianyu Chen**[*†]     **Kevin Bello**[‡§]     **Bryon Aragam**[‡]     **Pradeep Ravikumar**[§]
[†]Department of Statistics and Data Science, University of Texas at Austin
[‡]Booth School of Business, University of Chicago
[§]Machine Learning Department, Carnegie Mellon University

## Abstract

Structural causal models (SCMs) are widely used in various disciplines to represent causal relationships among variables in complex systems. Unfortunately, the underlying causal structure is often unknown, and estimating it from data remains a challenging task. In many situations, however, the end goal is to localize the changes (shifts) in the causal mechanisms between related datasets instead of learning the full causal structure of the individual datasets. Some applications include root cause analysis, analyzing gene regulatory network structure changes between healthy and cancerous individuals, or explaining distribution shifts. This paper focuses on identifying the causal mechanism shifts in two or more related datasets over the same set of variables—*without estimating the entire DAG structure of each SCM*. Prior work under this setting assumed linear models with Gaussian noises; instead, in this work we assume that each SCM belongs to the more general class of *nonlinear* additive noise models (ANMs). A key technical contribution of this work is to show that the Jacobian of the score function for the *mixture distribution* allows for the identification of shifts under general non-parametric functional mechanisms. Once the shifted variables are identified, we leverage recent work to estimate the structural differences, if any, for the shifted variables. Experiments on synthetic and real-world data are provided to showcase the applicability of this approach. Code implementing the proposed method is open-source and publicly available at https://github.com/kevinsbello/iSCAN.

## 1 Introduction

Structural causal models (SCMs) are powerful models for representing causal relationships among variables in a complex system [54, 58]. Every SCM has an underlying graphical structure that is generally assumed to be a directed acyclic graph (DAG). Identifying the DAG structure of an SCM is crucial since it enables reasoning about interventions [54]. Nonetheless, in most situations, scientists can only access *observational* or *interventional* data, or both, while the true underlying DAG structure remains *unknown*. As a result, in numerous disciplines such as computational biology [66, 30, 20], epidemiology [64], medicine [61, 62], and econometrics [34, 28, 18], it is critically important to develop methods that can estimate the entire underlying DAG structure based on available data. This task is commonly referred to as causal discovery or structure learning, for which a variety of algorithms have been proposed over the last decades.

Throughout this work, we make the assumption of causal sufficiency (i.e., non-existence of unobserved confounders). Under this condition alone, identifying the underlying DAG structure is not possible in general, and remains worst-case NP-complete [14, 16]. Indeed, prominent methods such as PC [71]

---

[*]Work done while at the Department of Statistics at the University of Chicago. Correspondence to kbello@cs.cmu.edu.

and GES [15] additionally require the arguably strong faithfulness assumption [77] to consistently estimate, in large samples, the Markov equivalent class of the underlying DAG. However, these methods are not consistent in high-dimensions unless one additionally assumes sparsity or small maximum-degree of the true DAG [36, 50, 78]. Consequently, the existence of hub nodes, which is a well-known feature in several networks [5, 6, 7], significantly complicates the DAG learning problem.

In many situations, however, the end goal is to *detect shifts (changes) in the causal mechanisms* between two (or more) related SCMs rather than recovering the *entire* underlying DAG structure of each SCM. For example, examining the mechanism changes in the gene regulatory network structure between healthy individuals and those with cancer may provide insights into the genetic factors contributing to the specific cancer; within biological pathways, genes could regulate various target gene groups depending on the cellular environment or the presence of particular disease conditions [32, 60]. In these examples, while the individual networks could be *dense*, the number of mechanism shifts could be *sparse* [69, 74, 55]. Finally, in root cause analysis, the goal is to identify the sources that originated observed changes in a joint distribution; this is precisely the setting we study in this work, where we model the joint distributions via SCMs, as also done in [52, 33].

In more detail, we focus on the problem of identifying mechanism shifts given datasets from two or more environments (SCMs) over the same observables. We assume that each SCM belongs to the class of additive noise models (ANMs) [29], i.e., each variable is defined as a nonlinear function over a subset of the remaining variables plus a random noise (see Section 2 for formal definitions). Importantly, we *do not* make any structural assumptions (e.g., sparsity, small maximum-degree, or bounded tree-width) on the individual DAGs. Even though ANMs are well-known to be identifiable [29, 59], we aim to detect the *local distribution changes* without estimating the full structures individually. See Figure 1 for a toy example of what we aim to estimate. A similar setting to this problem was studied in [82, 23] albeit in the restrictive linear setting. Finally, it is worth noting that even with *complete knowledge of the entire structure of each SCM*, assessing changes in *non-parametric* functions across different groups or environments remains a very challenging problem [see for instance, 44].

**Contributions.** Motivated by recent developments on causal structure learning of ANMs [65], we propose a two-fold algorithm that (1) Identifies shifted variables (i.e., variables for which their causal mechanism has changed across the environments); and (2) If needed, for each shifted variable, estimates the structural changes among the SCMs. More concretely, we make the following set of contributions:

- To identify shifted variables (Definition 3), we prove that the variance of the diagonal elements of the Hessian matrix associated with the log-density of the *mixture distribution* unveils information to detect distribution shifts in the leaves of the DAGs (see Theorem 1). Due to this result, our algorithm (Algorithm 1) iteratively chooses a particular leaf variable and determines whether or not such variable is shifted. Importantly, this detection step **does not** rely on any structural assumptions on the individual DAGs, and can consistently detect distribution shifts for *non-parametric functionals* under very mild conditions such as second-order differentiability.

- To identify structurally shifted edges (Definition 4), we propose a nonparametric local parents recovery method (Algorithm 2) based on a recent measure of conditional dependence [3]. In addition, based on recent results in [4], we provide a theoretical justification for the asymptotic consistency of Algorithm 2 in Theorem 2. Importantly, since structural changes can only occur on *shifted nodes*, this second step can be conducted much more efficiently when the sparse mechanism shift hypothesis [69] holds, which posits that only a small subset of the causal model's mechanisms change.

- We empirically demonstrate that our method can outperform existing methods such as DCI, which is tailored for linear models, as well as related methods for estimating unknown intervention targets such as UT-IGSP [72]. See Section 5 and Appendix C for more details. Moreover, in Section 5.2, we provide experiments on an ovarian cancer dataset, thus, showcasing the applicability of our method.

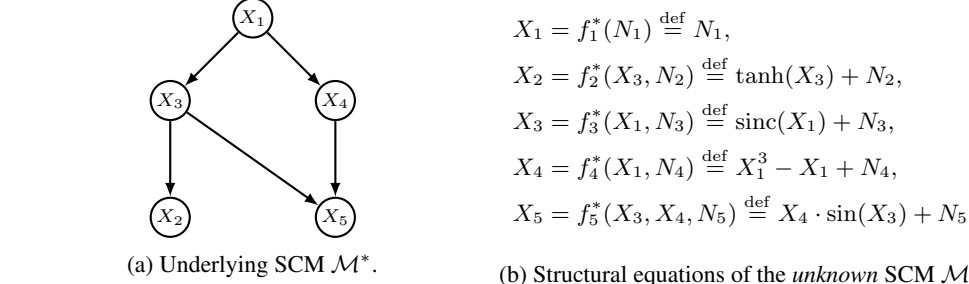

(a) Underlying SCM $\mathcal{M}^*$.

$$X_1 = f_1^*(N_1) \stackrel{\text{def}}{=} N_1,$$
$$X_2 = f_2^*(X_3, N_2) \stackrel{\text{def}}{=} \tanh(X_3) + N_2,$$
$$X_3 = f_3^*(X_1, N_3) \stackrel{\text{def}}{=} \text{sinc}(X_1) + N_3,$$
$$X_4 = f_4^*(X_1, N_4) \stackrel{\text{def}}{=} X_1^3 - X_1 + N_4,$$
$$X_5 = f_5^*(X_3, X_4, N_5) \stackrel{\text{def}}{=} X_4 \cdot \sin(X_3) + N_5.$$

(b) Structural equations of the *unknown* SCM $\mathcal{M}^*$.

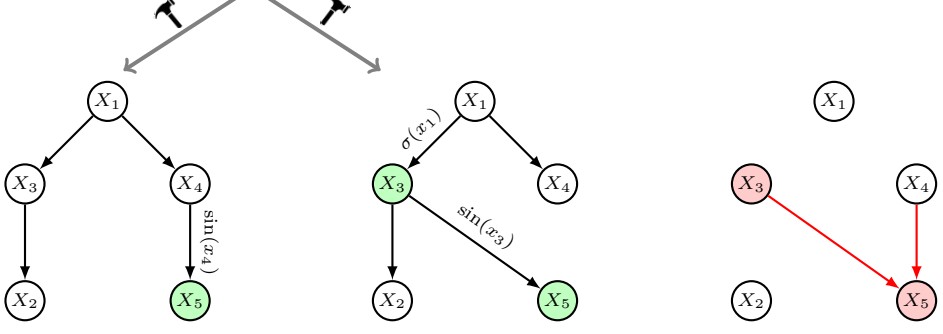

(c) Env. $\mathcal{E}_1$, originated by an *unknown* intervention on $X_5$.

(d) Env. $\mathcal{E}_2$, originated by *unknown* interventions on $X_3$ and $X_5$.

(e) Shifts in mechanisms between $\mathcal{E}_1$ and $\mathcal{E}_2$.

Figure 1: Illustration of two different environments (see Definition 2) in (1c) and (1d), both originated from the underlying SCM in (1a) with structural equations given in (1b). Between the two environments, we observe a change in the causal mechanisms of variables $X_3$ and $X_5$—the red nodes in (1e). Specifically, for $X_5$, we observe that its *functional dependence* changed from $X_4$ in $\mathcal{E}_1$ to $X_3$ in $\mathcal{E}_2$. For $X_3$, its *structural dependence* has not changed between $\mathcal{E}_1$ and $\mathcal{E}_2$, and only its functional changed from $\text{sinc}(X_1)$ in $\mathcal{E}_1$ to the sigmoid function $\sigma(X_1)$ in $\mathcal{E}_2$. Finally, in (1e), the red edges represent the *structural* changes in the mechanisms. The non-existence of an edge from $X_1$ to $X_3$ indicates that the structural relation between $X_1$ and $X_3$ is invariant.

## 2 Preliminaries and Background

In this section we introduce notation and formally define the problem setting. We use $[d]$ to denote the set of integers $\{1, \ldots, d\}$. Let $G = ([d], E)$ be a DAG with node set $[d]$ and a set of directed edges $E \subset [d] \times [d]$, where any $(i, j) \in E$ indicates and edge from $i$ to $j$. Also let $X = (X_1, \ldots, X_d)$ denote a $d$-dimensional vector of random variables. An SCM $\mathcal{M} = (X, f, \mathbb{P}_N)$ over $d$ variables is generally defined as a collection of $d$ structural equations of the form:

$$X_j = f_j(\text{PA}_j, N_j), \forall j \in [d], \tag{1}$$

where $\text{PA}_j \subseteq \{X_1, \ldots, X_d\} \setminus \{X_j\}$ are the *direct causes* (or parents) of $X_j$; $f = \{f_j\}_{j=1}^d$ is a set of *functional mechanisms* $f_j : \mathbb{R}^{|\text{PA}_j|+1} \to \mathbb{R}$; and $\mathbb{P}_N$ is a joint distribution[2] over the noise variables $N_j$, which we assume to be jointly independent[3]. Moreover, the underlying graph $G$ of an SCM is constructed by drawing directed edges for each $X_k \in \text{PA}_j$ to $X_j$. We henceforth assume this graph to be acyclic, i.e., a DAG. Finally, every SCM $\mathcal{M}$ defines a unique distribution $\mathbb{P}_X$ over the variables $X$ [Proposition 6.3 in 58], which by the independence of the noise variables (a.k.a. the Markovian assumption), $\mathbb{P}_X$ admits the following factorization:

$$\mathbb{P}(X) = \prod_{j=1}^d \mathbb{P}(X_j \mid \text{PA}_j), \tag{2}$$

where $\mathbb{P}(X_j \mid \text{PA}_j)$ is referred as the *causal mechanism* of $X_j$.

---

[2]We will always assume the existence of a density function w.r.t. the Lebesgue measure.

[3]Note that this implies that there is no hidden confounding.

The model above is often too general due to problems of identifiability. In this work we will consider that the noises are additive.

**Definition 1** (Additive noise models (ANMs)). *An additive noise model is an SCM $\mathcal{M} = (X, f, \mathbb{P}_N)$ as in* (1)*, where each structural assignment has the form:*

$$X_j = f_j(\mathrm{PA}_j) + N_j, \forall j \in [d].$$

Depending on the assumptions on $f_j$ and $N_j$, the underlying DAG of an ANM can be identifiable from observational data. E.g., when $f_j$ is linear and $N_j$ is Gaussian, in general one can only identify the Markov equivalence class (MEC) of the DAG, assuming faithfulness [54]. For linear models, an exception arises when assuming equal error variances [56, 78, 47], or non-Gaussian errors [70]. In addition, when $f_j$ is nonlinear on each component and three times differentiable then the DAG is also identifiable [59, 29]. Very recently, Rolland et al. [65] proved DAG identifiability when $f_j$ is nonlinear on each component and $N_j$ is Gaussian, using information from the score's Jacobian.

## 2.1 Data from multiple environments

Throughout this work we assume that we observe a collection of datasets, $\mathcal{D} = \{\boldsymbol{X}^h\}_{h=1}^H$, from $H$ (possibly different) environments. Each dataset $\boldsymbol{X}^h = \{X^{h,i}\}_{i=1}^{m_h}$ from environment $h$ contains $m_h$ (possibly non-independent) samples from the joint distribution $\mathbb{P}_X^h$, i.e., $\boldsymbol{X}^h \in \mathbb{R}^{m_h \times d}$. We consider that each environment originates from soft interventions[4] [54] of an *unknown* underlying SCM $\mathcal{M}^*$ with DAG structure $G^*$ and joint distribution $\mathbb{P}^*(X) = \prod_{j=1}^d \mathbb{P}^*(X_j \mid \mathrm{PA}_j^*)$. Here $\mathrm{PA}_j^*$ denotes the parents (direct causes) of $X_j$ in $G^*$. Then, an environment arises from manipulations or shifts in the causal mechanisms of a *subset* of variables, transforming from $\mathbb{P}^*(X_j \mid \mathrm{PA}_j^*)$ to $\widetilde{\mathbb{P}}(X_j \mid \widetilde{\mathrm{PA}}_j)$. Throughout, we will make the common modularity assumption of causal mechanisms [54, 38], which postulates that an intervention on a node $X_j$ only changes the mechanism $\mathbb{P}(X_j \mid \mathrm{PA}_j)$, while all other mechanisms $\mathbb{P}(X_i \mid \mathrm{PA}_i)$, for $i \neq j$, remain unchanged.

**Definition 2** (Environment). *An environment $\mathcal{E}_h = (X, f^h, \mathbb{P}_N^h)$, with joint distribution $\mathbb{P}_X^h$ and density $p_x^h$, independently results from an SCM $\mathcal{M}^*$ by intervening on an* unknown *subset $S^h \subseteq [d]$ of causal mechanisms, that is, we can factorize the joint distribution $\mathbb{P}^h(X)$ as follows:*

$$\mathbb{P}^h(X) = \prod_{j \in [d]} \mathbb{P}^h(X_j \mid \mathrm{PA}_j^h) = \prod_{j \in S^h} \widetilde{\mathbb{P}}^h(X_j \mid \widetilde{\mathrm{PA}}_j^h) \prod_{j \notin S^h} \mathbb{P}^*(X_j \mid \mathrm{PA}_j^*), \tag{3}$$

*where $\widetilde{\mathrm{PA}}_j^h$ is a (possibly empty) subset of the underlying causal parents $\mathrm{PA}_j^*$, i.e., $\widetilde{\mathrm{PA}}_j^h \subseteq \mathrm{PA}_j^*$; and, $\mathbb{P}^*(X_j \mid \mathrm{PA}_j^*)$ are the invariant mechanisms.*

**Remark 1.** *In the literature [e.g., 55], it is common to find the assumption that in a soft intervention the direct causes remain invariant, i.e., $\widetilde{\mathrm{PA}}_j^h = \mathrm{PA}_j^*$ for all $j \in S^h, h \in [H]$. In this work we consider a more general setting where none, some, or all of the direct causes of an intervened node are removed, i.e., $\widetilde{\mathrm{PA}}_j^h \subseteq \mathrm{PA}_j^*$ for all $j \in S^h, h \in [H]$.*

We next define shifted nodes (variables).

**Definition 3** (Shifted node). *Given $H$ environments $\{\mathcal{E}_h = (X, f^h, \mathbb{P}_N^h)\}_{h=1}^H$ originated from an ANM $\mathcal{M}^*$, a node $j$ is called a shifted node if there exists $h, h' \in [H]$ such that:*

$$\mathbb{P}^h(X_j \mid \mathrm{PA}_j^h) \neq \mathbb{P}^{h'}(X_j \mid \mathrm{PA}_j^{h'}).$$

To conclude this section, we formally define the problem setting.

**Problem setting.** Given $H$ datasets $\{\boldsymbol{X}^h\}_{h=1}^H$, where $\boldsymbol{X}^h \sim \mathbb{P}_X^h$ consists of $m_h$ (possibly non-independent) samples from the environment distribution $\mathbb{P}_X^h$ originated from an underlying ANM $\mathcal{M}^*$, estimate the set of shifted nodes and structural differences.

We note that [82, 23] have study the problem setting above for $H = 2$, assuming *linear functions* $f_j^h$, and Gaussian noises $N_j^h$. In this work, we consider a more challenging setting where $f_j^h$ are nonparametric functions (see Section 3 for more details).

---

[4]These types of interventions are more realistic in practice than "hard" or perfect interventions. However, note that we allow a soft intervention on a variable to remove some or all of its causes, where the latter is also known as an stochastic hard intervention.

## 2.2 Related Work

First we mention works most closely related to ours. The problem of learning the difference between *undirected* graphs has received much more attention than the directed case. E.g., [88, 46, 85, 19] develop algorithms for estimating the difference between Markov random fields and Ising models. See [87] for recent developments in this direction. In the directed setting, [82, 23] propose methods for directly estimating the difference of linear ANMs with Gaussian noise. More recently, [67] studied the setting where a dataset is generated from a mixture of SCMs, and their method is capable of detecting conditional distributions changes; however, due to the unknown membership of each sample, it is difficult to test for structural and functional changes. Moreover, in contrast to ours, all the aforementioned work on the directed setting rely on some form of faithfulness assumption.

**Causal discovery from a single environment.** One way to identify mechanism shifts (albeit inefficient) would be to estimate the individual DAGs for each environment and then test for structural differences across the different environments. A few classical and recent methods for learning DAGs from a single dataset include: Constraint-based algorithms such as PC and FCI [71]; in score-based methods, we have greedy approaches such as GES [16], likelihood-based methods [56, 47, 59, 2, 1, 29], and continuous-constrained learning [89, 51, 39, 8]. Order-based methods [75, 41, 24, 65, 48], methods that test for asymmetries [70, 12], and hybrid methods [50, 76]. Finally, note that even if we *perfectly estimate each individual DAG* (assuming identifiable models such as ANMs), applying these methods would only identify *structural* changes. That is, for variables that have the same parents across all the environments, we would require an additional step to identify *distributional* changes.

**Testing functional changes in multiple datasets.** Given the parents of a variable $X_j$, one could leverage prior work [44, 25, 9, 26] on detecting heterogeneous functional relationships. However, we highlight some important limitations. Several methods such as [25, 9, 26] only work for one dimensional functionals and assume that the datasets share the exact same design matrix. Although [44] relaxes this assumption and extends the method to multivariate cases, the authors assume that the covariates (i.e., $\mathrm{PA}_j^h$) are sampled from the *same distribution* across the environments, which is a strong assumption in our context since ancestors of $X_j$ could have experienced mechanism shifts. Finally, methods such as [53] and [11], although nonparametric, need knowledge about the parent set $\mathrm{PA}_j$ for each variable, and they assume that $\mathrm{PA}_j$ is same across different environments.

**Causal discovery from heterogeneous data.** Another well-studied problem is to learn the underlying DAG of the SCM $\mathcal{M}^*$ that originated the different environments. Under this setting, [83] provided a characterization of the $\mathcal{I}$-MEC, a subset of the Markov equivalence class. [55] provided DAG-identifiability results by leveraging sparse mechanism shifts and relies on identifying such shifts, which this work aims to solve. [10] developed an estimator considering unknown intervention targets. [79] primarily focuses on linear SEM and does not adapt well to nonlinear scenarios. Also assuming linear models, [22, 21] applied ideas from linear invariant causal prediction [ICP, 57] and ICM to identify the causal DAG. [72] proposes a nonparametric method that can identify the intervention targets; however, this method relies on nonparametric CI tests, which can be time-consuming and sample inefficient. [49] introduced the joint causal inference (JCI) framework, which can also estimate intervention nodes. However, this method relies on an assumption that the intervention variables are fully connected, a condition that is unlikely to hold in practice. [31] introduced a two-stage approach that removes functional restrictions. First, they used the PC algorithm using all available data to identify the MEC. Then, the second step aims to orient the remaining edges based on a novel measure of mechanism dependence. Finally, we note that a common assumption in the aforementioned methods is the knowledge of which dataset corresponds to the observational distribution; without such information, their assumptions on the type of interventions would not hold true. In contrast, our method does not require knowledge of the observational distribution.

## 3 Identifying Causal Mechanism Shifts via Score Matching

In this section, we propose iSCAN (*identifying Shifts in Causal Additive Noise models*), a method for detecting shifted nodes (Definition 3) based only on information from the Jacobian of the score of the data distribution[5].

---

[5]In this work, the score of a pdf $p(x)$ means $\nabla \log p(x)$

Let $X$ be the row concatenation of all the datasets $X^h$, i.e., $X = [(X^1)^\top \mid \cdots \mid (X^H)^\top]^\top \in \mathbb{R}^{m \times d}$, where $m = \sum_{h=1}^H m_h$. The pooled data $X$ can be interpreted as a mixture of data from the $H$ different environments. To account for this mixture, we introduce the probability mass $w_h$, which represents the probability that an observation belongs to environment $h$, i.e., $\sum_{h=1}^H w_h = 1$. Let $\mathbb{Q}(X)$ denote the distribution of the mixture data with density function $q(x)$, i.e., $q(x) = \sum_{h=1}^H w_h p^h(x)$.

In the sequel, we use $s^h(x) \equiv \nabla \log p^h(x)$ to denote the score function of the joint distribution of environment $h$ with density $p^h(x)$. Also, we let $s(x) \equiv \nabla \log q(x)$ to denote the score function of the mixture distribution with density $q(x)$. We will make the following assumptions on $f_j^h$ and $N_j^h$.

**Assumption A.** *For all $h \in [H], j \in [d]$, the functional mechanisms $f_j^h(\mathrm{PA}_j^h)$ are assumed to be non-linear in every component.*

**Assumption B.** *For all $j \in [d], h \in [H]$, the pdf of the real-valued noise $N_j^h$ denoted by $p_{N_j}^h$ satisfies $\frac{\partial^2}{(\partial n_j^h)^2} \log p_{N_j}^h(n_j^h) = c_j^h$ where $c_j^h$ is a non-zero constant. Moreover, $\mathbb{E}[N_j^h] = 0$.*

For an ANM, Rolland et al. [65] showed that under Assumption A and assuming zero-mean Gaussian noises (which satisfies Assumption B), the diagonal of the Jacobian of the score function reveals the leaves of the underlying DAG. We next instantiate their result in our context.

**Proposition 1** (Lemma 1 in [65, 68])**.** *For an environment $\mathcal{E}_h$ with underlying DAG $G^h$ and pdf $p^h(x)$, let $s^h(x) = \nabla \log p^h(x)$ be the associated score function. Then, under Assumptions A and B, for all $j \in [d]$, we have:*

$$\text{Node } j \text{ is a leaf in } G^h \iff \mathrm{Var}_X\left[\frac{\partial s_j^h(X)}{\partial x_j}\right] = 0.$$

Motivated by the ideas of leaf-identifiability from the score's Jacobian in a *single* ANM, we next show that the *score's Jacobian of the mixture distribution* can help reveal mechanism shifts among the different environments.

**Theorem 1.** *For all $h \in [H]$, let $G^h$ and $p^h(x)$ denote the underlying DAG structure and pdf of environment $\mathcal{E}_h$, respectively, and let $q(x)$ be the pdf of the mixture distribution of the $H$ environments such that $q(x) = \sum_{h=1}^H w_h p^h(x)$. Also, let $s(x) = \nabla \log q(x)$ be the associated score function. Then, under Assumptions A, and B, we have:*

*(i) If $j$ is a leaf in all DAGs $G^h$, then $j$ is a shifted node if and only if $\mathrm{Var}_X\left[\frac{\partial s_j(X)}{\partial x_j}\right] > 0$.*

*(ii) If $j$ is not a leaf in at least one DAG $G^h$, then $\mathrm{Var}_X\left[\frac{\partial s_j(X)}{\partial x_j}\right] > 0$.*

Theorem 1 along with Proposition 1 suggests a way to identify shifted nodes. Namely, to use Proposition 1 to identify a common leaf, and then use Theorem 1 to test if such a leaf is a shifted node or not. We then proceed to remove the leaf and repeat the process. See Algorithm 1. Note that due to the fact that each environment is a result of an intervention (Definition 2) on an underlying ANM $\mathcal{M}^*$, it follows that the leaves in $G^*$ will remain leaves in each DAG $G^h$.

---

**Algorithm 1 iSCAN**—Identifying Shifts in Causal Additive Noise models.

---

**Input:** Datasets $X^1, \ldots, X^H$.
**Output:** Shifted variables set $\widehat{S}$, and topological sort $\hat{\pi}$.
1: Initialize $\widehat{S} = \{\}, \hat{\pi} = (\ ), \mathcal{N} = \{1, \ldots, d\}$
2: Set $X = [(X^1)^\top \mid \cdots \mid (X^H)^\top]^\top \in \mathbb{R}^{m \times d}$.
3: **while** $\mathcal{N} \neq \emptyset$ **do**
4:      $\forall h \in [H], \mathrm{Var}^h \leftarrow \mathrm{Var}_{X^h}\left[\mathrm{diag}(\nabla^2 \log p^h(x))\right]$.
5:      $\mathrm{Var} \leftarrow \mathrm{Var}_X\left[\mathrm{diag}(\nabla^2 \log q(x))\right]$
6:      $L \leftarrow \bigcap_{h \in [H]}\left\{j \mid \mathrm{Var}_j^h = 0, j \in [d]\right\}$.            ▷ Identify leaves.
7:      $\widehat{S} \leftarrow \widehat{S} \bigcup \left\{j \mid \mathrm{Var}_j \neq 0, j \in L\right\}$            ▷ Identify shifted nodes.
8:      $\mathcal{N} \leftarrow \mathcal{N} - \{L\}$
9:      $\forall l \in L$, remove the $l$-th column of $X^h, \forall h \in [H]$, and $X$.
10:     $\hat{\pi} \leftarrow (L, \hat{\pi})$.

---

**Remark 2.** *See Appendix A for a practical implementation of Alg. 1. Finally, note that Alg. 1 also estimates a valid topological sort for the different environments by leveraging Proposition 1.*

### 3.1 Score's Jacobian estimation

Since the procedure to estimate $\mathrm{Var}_q[\frac{\partial s_j(x)}{\partial x_j}]$ is similar for estimating $\mathrm{Var}_{p^h}[\frac{\partial s_j^h(x)}{\partial x_j}]$ for each $h \in [H]$, in this section we discuss the estimation for $\mathrm{Var}_q[\frac{\partial s_j(x)}{\partial x_j}]$, which involves computing the diagonal of the Hessian of $\log q(x)$. To estimate this quantity, we adopt a similar approach to the method in [45, 65]. First, we estimate the first-order derivative of $\log q(x)$ by Stein's identity [73]:

$$\mathbb{E}_q\left[\boldsymbol{h}(x)\nabla \log q(x)^\top + \nabla \boldsymbol{h}(x)\right] = 0, \tag{4}$$

where $\boldsymbol{h}: \mathbb{R}^d \to \mathbb{R}^{d'}$ is any test function such that $\lim_{x\to\infty} \boldsymbol{h}(x)q(x) = 0$. Once we have estimated $\nabla \log q(x)$, we can proceed to estimate the Hessian's diagonal by using second-order Stein's identity:

$$\mathbb{E}_q[\boldsymbol{h}(x)\mathrm{diag}(\nabla^2 \log q(x))^\top] = \mathbb{E}_q[\nabla^2_{\mathrm{diag}}\boldsymbol{h}(x) - \boldsymbol{h}(x)\mathrm{diag}(\nabla \log q(x)\nabla \log q(x)^\top)] \tag{5}$$

Using eq.(4) and eq.(5), we can estimate the Hessian's diagonal at each data point. Thus allowing us to obtain an estimate of $\mathrm{Var}_q\left[\frac{\partial s_j(x)}{\partial x_j}\right]$. See Appendix A.1 for additional details.

**Remark 3** (Consistency of Algorithm 1). *The estimators in eq.(6) and eq.(7), given in Appendix A.1, correspond to Monte Carlo estimators using eq.(4) and (5), respectively, then the error of the estimators tend to zero as the number of samples goes to infinity. See for instance the discussion in Section 3.1 in [45]. We empirically explore the consistency of Algorithm 1 in Figure 2.*

**Remark 4** (Computational Complexity). *Since we adopt the kernel-based estimator, SCORE, from [65]. The computational complexity for the estimation of the score's Jacobian in a single environment is $\mathcal{O}(dm_h^3)$. In Algorithm 1, computation is dominated by the SCORE function applied to the pooled data $\boldsymbol{X} \in \mathbb{R}^{m\times d}$. Therefore, the overall complexity of Algorithm 1 is $\mathcal{O}(dm^3)$. See Figure 2.*

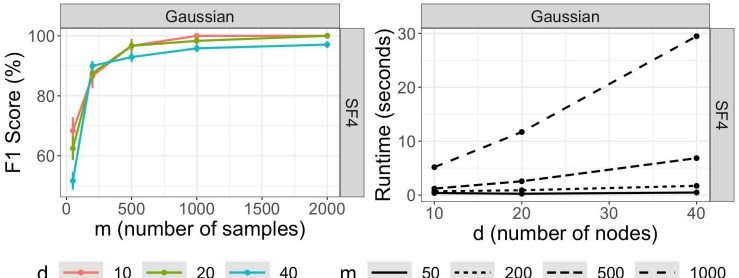

Figure 2: (Left) F1 score of the output of Alg. 1 w.r.t. to the true set of shifted nodes. For different number of nodes, we observe how iSCAN recovers the true set of shifted nodes as the number of samples increases, thus empirically showing its consistency. (Right) Runtime vs number of nodes for different number of samples. We corroborate the linear dependence of the time complexity on $d$.

## 4 On Identifying Structural Differences

After estimating the set of shifted nodes $\widehat{S}$ through Algorithm 1, it is of high interest to predict which causal relations between a shifted node and its parents have undergone changes across the environments. The meaning of a change in a causal relationship can vary based on the context and the estimation objective. This section primarily centers on structural changes, elaborated further below, while additional discussion about other types of changes is available in Appendix D.

**Definition 4** (Structurally shifted edge). *For a given shifted node $X_j$, an edge $X_i \to X_j$ is called a structurally shifted edge if $\exists h, h' \in [H]$ such that $X_i \in \mathrm{PA}_j^h$ and $X_i \notin \mathrm{PA}_j^{h'}$.*

In other words, a structurally shifted edge is an edge that exists in one environment but not in another, indicating a change in the underlying structure of the causal mechanism. To detect the structurally shifted edges, we will estimate the parents of each shifted node in $\widehat{S}$ for all environments $\mathcal{E}_h$.

**Remark 5.** *Note that under the sparse mechanism shift hypothesis [69], i.e., $|S| \ll d$, estimating the parents of each shifted node is much more efficient than estimating the entire individual structures.*

**Kernel regression and variable selection.** A potential strategy to estimate structurally shifted edges involves employing the estimated topological order $\hat{\pi}$ obtained from Algorithm 1. If this estimated topological order remains valid across all environments, it can serve as a guide for the nonparametric variable selection process to identify the parents of a shifted node $X_j$. Specifically, we can regress the shifted node $X_j$ on its predecessors $\widehat{\mathrm{Pre}}(X_j)$ and proceed with a nonparametric variable selection procedure. Here $\widehat{\mathrm{Pre}}(X_j)$ consists of the set of nodes that appear before $X_j$ in the estimated topological order $\hat{\pi}$. To achieve that, there exist various methods under the hypothesis testing framework [42, 17, 63], and bandwidth selection procedures [40]. These methods offer consistency guarantees, but their time complexity might be problematic. Kernel regression, for example, has a time complexity of $\mathcal{O}(m^3)$, and requires an additional bandwidth selection procedure, usually with a time complexity of $\mathcal{O}(m^2)$. Consequently, it becomes imperative to find a more efficient method for identifying parents locally.

**Feature ordering by conditional independence (FOCI).** An alternative efficient approach for identifying the parents is to leverage the feature ordering method based on conditional independence proposed by Azadkia and Chatterjee [3]. This method provides a measure of conditional dependency between variables with a time complexity of $\mathcal{O}(m \log m)$. By applying this method, we can perform fast variable selection in a nonparametric setting. See Algorithm 4 in Appendix A.3.

**Theorem 2** (Consistency of Algorithm 4). *Under Assumption C, given in Appendix B.2, if the estimated topological order $\hat{\pi}$ output from Algorithm 1 is valid for all environments, then the output $\widehat{\mathrm{PA}}_j^h$ of Algorithm 4 is equal to the true parents $\mathrm{PA}_j^h$ of node $X_j$ with high probability, for all $h \in [H]$.*

Motivated by Theorem 2, we next present Algorithm 2, a procedure to estimate the structurally shifted edges. Given the consistency of Alg. 1 and Alg. 2, it follows that combining both algorithms will correctly estimate the true set of shifted nodes and structural shifted edges, asymptotically.

---

**Algorithm 2** Identifying structurally shifted edges

---

**Input:** Data $\{\boldsymbol{X}^h\}_{h \in [H]}$, topological order $\hat{\pi}$, shifted nodes $\widehat{S}$
**Output:** Structurally shifted edges set $\widehat{E}$
 1: Initialize $\widehat{E} = \emptyset$
 2: **for** $X_j$ in $\widehat{S}$ **do**
 3:     **for** $h$ in $[H]$ **do**
 4:         Estimate $\widehat{\mathrm{PA}}_j^h$ from Alg. 4 (FOCI) with input $\{\widehat{\mathrm{Pre}}(\boldsymbol{X}_j^h), \boldsymbol{X}_j^h\}$
 5:     **if** $\exists X_k, h, h'$ such that $X_k \in \widehat{\mathrm{PA}}_j^h, X_k \notin \widehat{\mathrm{PA}}_j^{h'}$ **then**
 6:         $\widehat{E} \leftarrow \widehat{E} \cup (X_k, X_j)$

---

## 5   Experiments

We conducted a comprehensive evaluation of our algorithms. Section 5.1 focuses on assessing the performance of iSCAN (Alg. 1) for identifying shifted variables. In Section 5.2, we apply iSCAN for identifying shifted nodes along with FOCI (Alg. 2) for estimating structural changes, on apoptosis data. Also, in App. C, we provide additional experiments including: *(i)* Localizing shifted nodes without structural changes (App. C.1), and where the functionals are sampled from Gaussian processes (App. C.1.1); *(ii)* Localizing shifted nodes and estimating structural changes when the underlying graphs are different; and *(iii)* Evaluating iSCAN using the elbow method for selecting shifted nodes (see App. C.3 and Remark 6). Code is publicly available at https://github.com/kevinsbello/iSCAN.

### 5.1   Synthetic experiments on shifted nodes

**Graph models.** We generated random graphs using the Erdős-Rényi (ER) and scale free (SF) models. For a given number of variables $d$, ER$k$ and SF$k$ indicate an average number of edges equal to $kd$.

**Data generation.** We first sampled a DAG, $G^1$, of $d$ nodes according to either the ER or SF model for env. $\mathcal{E}_1$. For env. $\mathcal{E}_2$, we initialized its DAG structure from env. $\mathcal{E}_1$ and produced structural changes by randomly selecting $0.2 \cdot d$ nodes from the non-root nodes. This set of selected nodes $S$, with cardinality $|S| = 0.2d$, correspond to the set of "shifted nodes". In env. $\mathcal{E}_2$, for each shifted node $X_j \in S$, we uniformly at random deleted at most three of its incoming edges, and use $D_j$ to denote the parents whose edges to $X_j$ were deleted; thus, the DAG $G^2$ is a subgraph of $G^1$. Then, in $\mathcal{E}_1$, each $X_j$ was defined as follows:

$$X_j = \sum_{i \in \mathrm{PA}_j^1 \setminus D_j} \sin(X_i^2) + \sum_{i \in D_j} 4\cos(2X_i^2 - 3X_i) + N_j$$

In $\mathcal{E}_2$, each $X_j$ was defined as follows:

$$X_j = \sum_{i \in \mathrm{PA}_j^2} \sin(X_i^2) + N_j$$

**Experiment details.** For each simulation, we generated 500 data points per environment, i.e., $m_1 = 500, m_2 = 500$ and $m = 1000$. The noise variances were set to 1. We conducted 30 simulations for each combination of graph type (ER or SF), noise type (Gaussian, Gumbel, and Laplace), and number of nodes ($d \in \{10, 20, 30, 50\}$). The running time was recorded by executing the experiments on an Intel Xeon Gold 6248R Processor with 8 cores. For our method, we used $\eta = 0.05$ for eq.(6) and eq.(7), and a threshold $t = 2$ (see Alg. 3).

**Evaluation.** We compared the performance of iSCAN against several baselines, which include: DCI [82], the approach by [11], CITE [79], KCD [53], SCORE [65], and UT-IGSP [72]. Figure 3 illustrates the results for ER4 and SF4 graphs. We note that iSCAN consistently outperforms other baselines in terms of F1 score across all scenarios. Importantly, note how the performance of some baselines, like DCI, CITE, Budhathoki's, and SCORE, degrades faster for graphs with hub nodes, a property of SF graphs. In contrast, iSCAN performs similarly, as it is not dependent on structural assumptions on the individual DAGs. Additionally, it is worth noting that our method exhibits faster computational time than KCD, Budhathoki's, and SCORE, particularly for larger numbers of nodes.

In Appendix C.1, we provide experiments on sparser graphs such as ER2/SF2, and denser graphs such as ER6/SF6. We also include Precision and Recall in all plots in the supplement.

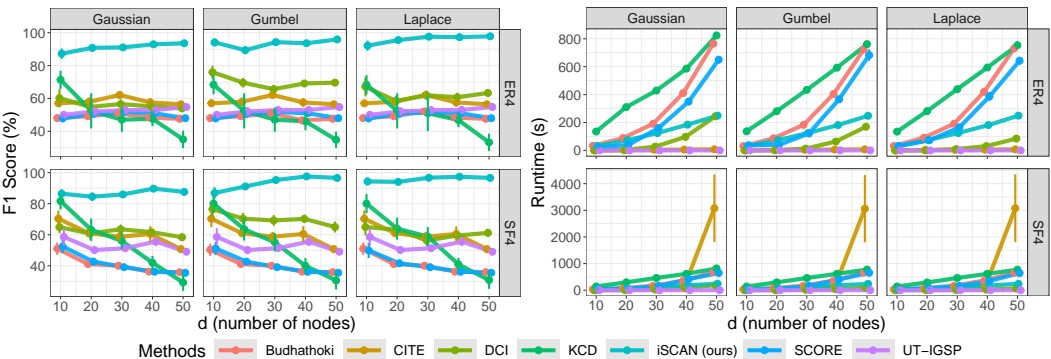

Figure 3: Experiments on ER4 and SF4 graphs. See the experiment details above. The points indicate the average values obtained from these simulations. The error bars depict the standard errors. Our method iSCAN (light blue) consistently outperformed baseline methods in terms of F1 score.

## 5.2 Experiments on apoptosis data

We conducted an analysis on an ovarian cancer dataset using iSCAN (Algorithm 1) to identify shifted nodes and Algorithm 2 to detect structurally shifted edges (SSEs). This dataset had previously been analyzed using the DPM method [88] in the undirected setting, and the DCI method [82] in the linear setting. By applying our method, we were able to identify the shifted nodes and SSEs in the dataset (see Figure 4a). Our analysis revealed the identification of two hub nodes in the apoptosis

pathway: BIRC3, and PRKAR2B. The identification of BIRC3 as a hub node was consistent with the results obtained by the DPM and DCI methods. Additionally, our analysis also identified PRKAR2B as a hub node, which was consistent with the result obtained by the DCI method. Indeed, BIRC3, in addition to its role in inhibiting TRAIL-induced apoptosis, has been investigated as a potential therapeutic target in cancer treatment including ovarian cancer[35, 81]; whereas PRKAR2B has been identified as an important factor in the progression of ovarian cancer cells. The latter serves as a key regulatory unit involved in the growth and development of cancer cells [84, 13].

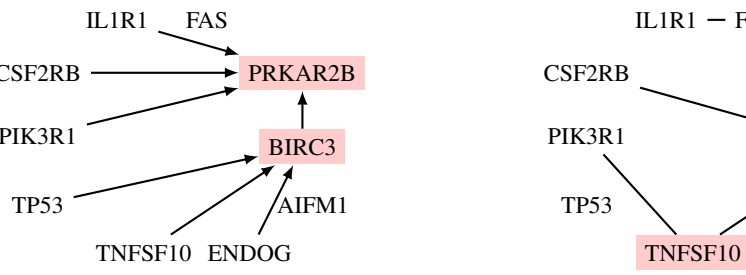

(a) The red nodes are the shifted nodes estimated by iSCAN (Alg. 1). The edges are the structurally shifted edges estimated by FOCI (Alg. 2).

(b) Undirected difference network estimated by DPM [88]. The red nodes indicate hub nodes, however, it is not clear which node mechanisms have changed.

Figure 4: Results on apoptosis data.

# 6 Conclusion

In this work, we showed a novel connection between score matching and identifying causal mechanism shifts among related heterogeneous datasets. This finding opens up a new and promising application for score function estimation techniques.

Our proposed technique consists of three modules. The first module evaluates the Jacobian of the score under the *individual* distributions and the *mixture* distribution. The second module identifies shifted features (variables) using the estimated Jacobians, allowing us to pinpoint the nodes that have undergone a mechanism shift. Finally, the third module aims to estimate structurally shifted edges, a.k.a. the difference DAG, by leveraging the information from the identified shifted nodes and the estimated topological order. *It is important to note that our identifiability result in Theorem 1 is agnostic to the choice of the score estimator.*

The strength of our result lies in its capability to recover the difference DAG in non-linear Additive Noise Models (ANMs) without making any assumptions about the parametric form of the functions or statistical independencies. This makes our method applicable in a wide range of scenarios where non-linear relationships and shifts in mechanisms are present.

## 6.1 Limitations and future work

While our work demonstrates the applicability of score matching in identifying causal mechanism shifts in the context of nonlinear ANMs, there are several limitations and areas for future exploration:

*Extension to other families of SCMs*: Currently, our method is primarily focused on ANMs where the noise distribution satisfies Assumption B, e.g., Gaussian distributions. It would be valuable to investigate the application of score matching in identifying causal mechanism shifts in other types of SCMs. Recent literature, such as [48], has extended score matching to additive Models with arbitrary noise for finding the topological order. Expanding our method to accommodate different noise models would enhance its applicability to a wider range of real-world scenarios.

*Convergence rate analysis*: Although the score matching estimator is asymptotically consistent, the convergence rate remains unknown in general. Understanding the convergence properties of the estimator is crucial for determining the sample efficiency and estimating the required number of samples to control the estimation error within a desired threshold. Further theoretical developments, such as [37], on score matching estimators would provide valuable insights into the performance and sample requirements of iSCAN.

## Acknowledgments and Disclosure of Funding

K.B. was supported by NSF under Grant # 2127309 to the Computing Research Association for the CIFellows 2021 Project. B.A. was supported by NSF IIS-1956330, NIH R01GM140467, and the Robert H. Topel Faculty Research Fund at the University of Chicago Booth School of Business. P.R. was supported by ONR via N000141812861, and NSF via IIS-1909816, IIS-1955532, IIS-2211907. We are also grateful for the support of the University of Chicago Research Computing Center for assistance with the calculations carried out in this work.

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

# SUPPLEMENTARY MATERIAL
# iSCAN: Identifying Causal Mechanism Shifts among Nonlinear Additive Noise Models

## A   Practical Implementation

In this section, we present a more practical version of Alg. 1 that considers estimation errors, see Alg. 3. First, we provide more details of the score's Jacobian estimation.

### A.1   Practical Version of SCORE

Let $\boldsymbol{X} = \{x^1, \ldots, x^m\}$ be a dataset of $m$ possibly non-independent but identically distributed samples. From Li and Turner [45], we next present the estimator for the point-wise first-order partial derivative, corresponding to eq.(4):

$$\hat{\boldsymbol{G}} = -(\boldsymbol{K} + \eta \boldsymbol{I})^{-1}\langle \nabla, \boldsymbol{K} \rangle \tag{6}$$

where $\boldsymbol{H} = (h(x^1), \ldots, h(x^m)) \in \mathbb{R}^{d' \times m}$, $\overline{\nabla \boldsymbol{h}} = \frac{1}{m}\sum_{k=1}^m \nabla \boldsymbol{h}(x^k)$, $\boldsymbol{K} = \boldsymbol{H}^\top \boldsymbol{H}$, $K_{ij} = \kappa(x^i, x^j) = \boldsymbol{h}(x^i)^\top \boldsymbol{h}(x^j)$, $\langle \nabla, \boldsymbol{K} \rangle = m\boldsymbol{H}^T \overline{\nabla \boldsymbol{h}}$, $\langle \nabla, \boldsymbol{K} \rangle_{ij} = \sum_{k=1}^m \nabla_{x_j^k} \kappa(x^i, x^j)$, and $\eta \geq 0$ is a regularization parameter. Here $\hat{\boldsymbol{G}}$ is used to approximate $\boldsymbol{G} \equiv (\nabla \log p(x^1), \ldots, \nabla \log p(x^m))^\top \in \mathbb{R}^{m \times d}$.

From [65], we now present the estimator for the diagonal elements of the score's Jacobian at the sample points, i.e. $\boldsymbol{J} \equiv (\mathrm{diag}(\nabla^2 \log p(x^1)), \ldots (\mathrm{diag}(\nabla^2 \log p(x^m)))^\top \in \mathbb{R}^{m \times d}$, the estimator of $\boldsymbol{J}$ is:

$$\hat{\boldsymbol{J}} = -\mathrm{diag}\left(\hat{\boldsymbol{G}}\hat{\boldsymbol{G}}^\top\right) + (\boldsymbol{K} + \eta \boldsymbol{I})^{-1}\langle \nabla_{\mathrm{diag}}^2, \boldsymbol{K} \rangle \tag{7}$$

where $\boldsymbol{H} = (h(x^1), \ldots, h(x^m)) \in \mathbb{R}^{d' \times m}$, $\overline{\nabla_{\mathrm{diag}}^2 \boldsymbol{h}} = \frac{1}{m}\sum_{k=1}^m \nabla_{\mathrm{diag}}^2 \boldsymbol{h}(x^k)$, $(\nabla_{\mathrm{diag}}^2 \boldsymbol{h}(x))_{ij} = \frac{\partial^2 h_i(x)}{\partial x_j^2}$, $\boldsymbol{K} = \boldsymbol{H}^\top \boldsymbol{H}$, $\boldsymbol{K}_{ij} = \kappa(x^i, x^j) = \boldsymbol{h}(x^i)^\top \boldsymbol{h}(x^j)$, $\langle \nabla_{\mathrm{diag}}^2, \boldsymbol{K} \rangle = m\boldsymbol{H}^T \overline{\nabla_{\mathrm{diag}}^2 \boldsymbol{h}}$, $\langle \nabla_{\mathrm{diag}}^2, \boldsymbol{K} \rangle_{ij} = \sum_{k=1}^m \frac{\partial^2 \kappa(x^i, x^k)}{(\partial x_j^k)^2}$, and $\eta \geq 0$ is a regularization parameter.

In the sequel, we use $\mathsf{SCORE}(\boldsymbol{X})$ to denote the procedure to compute the sample variance for the estimator of the diagonal of the score's Jacobian via eq.(7).

### A.2   Practical Version of Algorithm 1

Let $\widehat{\mathrm{Var}}^h$ be a $d$-dimensional vector, where $d$ is the number of nodes. We introduce a $d$-dimensional vector $\mathtt{rank}^h$, which represents the index of each element in $\widehat{\mathrm{Var}}^h$ after a non-decreasing sorting. For example, if $\widehat{\mathrm{Var}}^h = (5.2, 3.1, 4.5, 1.6)$, then $\mathtt{rank}^h = (3, 1, 2, 0)$. Furthermore, we define a $d$-dimensional vector $\mathtt{rank}$ as the element-wise summation of $\mathtt{rank}^h$ over all $h \in [H]$. In other words, $\mathtt{rank}$ is calculated as $\mathtt{rank} = \sum_{h \in [H]} \mathtt{rank}^h$.

Recall that in Section 3.1 we remarked that we leverage the SCORE approach from Rolland et al. [65] for estimating $\mathrm{diag}(\nabla^2 \log p(x))$ at each data point. Recall also that our identifiability result (Theorem 1) depends on determining whether a leaf node has variance $\mathrm{Var}_q(\frac{\partial s_j(x)}{\partial x_j}) = 0$. In practice, it is unrealistic to simply test for the equality $\mathrm{Var}_L = 0$ since $\mathrm{Var}_L$ carries out errors due to finite samples. Instead, we define the following statistic for each estimated leaf node $L$ (Line 10 in Algorithm 3):

$$\mathtt{stats}_L = \frac{\mathrm{Var}_L}{\min_h \mathrm{Var}_L^h + \epsilon}. \tag{8}$$

---

**Algorithm 3** Practical version of Algorithm 1

---

**Input:** Datasets $\boldsymbol{X}^1, \ldots, \boldsymbol{X}^H$, threshold $t$
**Output:** Shifted variables set $\widehat{S}$, and topological sort $\hat{\pi}$.

1: Initialize $\widehat{S} = \emptyset, \hat{\pi} = (\,), \mathcal{N} = \{1, \ldots, d\}$
2: $\texttt{stats} \leftarrow (0, \ldots, 0) \in \mathbb{R}^d$
3: Set $\boldsymbol{X} = [(\boldsymbol{X}^1)^\top \mid \cdots \mid (\boldsymbol{X}^H)^\top]^\top \in \mathbb{R}^{m \times d}$.
4: **while** $\mathcal{N} \neq \emptyset$ **do**
5:      $\forall h \in [H], \widehat{\mathrm{Var}}^h \leftarrow \mathsf{SCORE}(\boldsymbol{X}^h).$            $\triangleright$ Estimate $\mathrm{Var}_{\boldsymbol{X}^h}\big[\mathrm{diag}(\nabla^2 \log p^h(x))\big]$.
6:      $\forall h \in [H], \texttt{rank}^h \leftarrow \arg\mathrm{sort}(\widehat{\mathrm{Var}}^h).$
7:      $\texttt{rank} \leftarrow \sum_{h \in [H]} \texttt{rank}^h$
8:      $\widehat{L} \leftarrow \arg\min_j \texttt{rank}_j$            $\triangleright$ Estimate a leaf node
9:      $\widehat{\mathrm{Var}} \leftarrow \mathsf{SCORE}(\boldsymbol{X})$            $\triangleright$ Estimate $\mathrm{Var}_{\boldsymbol{X}}\big[\mathrm{diag}(\nabla^2 \log q(x))\big]$.
10:      $\texttt{stats}_L = \dfrac{\widehat{\mathrm{Var}}_L}{\min_h \widehat{\mathrm{Var}}_L^h}$
11:      $\mathcal{N} \leftarrow \mathcal{N} - \{L\}$
12:      Remove the $\widehat{L}$-th column of $\boldsymbol{X}^h, \forall h \in [H]$, and $\boldsymbol{X}$.
13:      $\hat{\pi} \leftarrow (\widehat{L}, \hat{\pi}).$
14: $\widehat{S} = \big\{ j \mid \texttt{stats}_j > t, \forall j \in [d] \big\}$

---

The intuition behind this ratio is that if the leaf node $L$ is a *shifted node* then we can expect $\frac{\mathrm{Var}_L}{\min_h \mathrm{Var}_L^h}$ to be large since $\mathrm{Var}_L > 0$ (by Theorem 1), and $\mathrm{Var}_L^h \approx 0$ (by Proposition 1). On the other hand, if the leaf node $L$ is **not** a shifted node then we can expect $\frac{\mathrm{Var}_L}{\min_h \mathrm{Var}_L^h}$ to be small. This is due to the fact that, given a consistent estimator, $\mathrm{Var}_L$ would converge towards 0 (by Theorem 1) at a faster rate than $\mathrm{Var}_L^h$ since we utilize a larger amount of data for estimating $\mathrm{Var}_L$. Finally, $\epsilon$ in the denominator is a very small value, e.g. $10^{-9}$, and acts as a safeguard against encountering a division by zero[6].

Then, given the statistic in eq.(8), we can set a threshold $t$ and define the set of shifted nodes $S$ by all the nodes $j$ such that $\texttt{stats}_j > t$ (Line 14 in Algorithm 3).

**Remark 6** (The elbow strategy). *Alternatively, we can employ an adaptive approach to identify the set of shifted nodes by sorting* $\texttt{stats}$ *in non-increasing order, and look for the "elbow" point. For example, Figure 5 illustrates the variance ratio in (8) for each node sorted in non-increasing order. In this case, node index 5 corresponds to the elbow point, allowing us to estimate nodes 5 and 8 as shifted nodes. Identifying the elbow point has the advantage to detect shifted nodes without relying on a fixed threshold.*

## A.3   Algorithm details for FOCI

---

**Algorithm 4** Feature ordering by conditional independence (FOCI)

---

**Input:** Data $\widehat{\mathrm{Pre}}(\boldsymbol{X}_j^h), \boldsymbol{X}_j^h$
**Output:** Estimated parents of $X_j, \widehat{\mathrm{PA}}_j^h$

1: $P \leftarrow \emptyset$
2: Let $T_m(i, j, P) \equiv T_m(\boldsymbol{X}_j^h, \boldsymbol{X}_i^h \mid \boldsymbol{X}_P^h)$      $\triangleright$ $T_m$ is the estimator in Azadkia and Chatterjee [3].
3: **while** $\max_{X_i \notin P, \, X_i \in \widehat{\mathrm{Pre}}(X_j)} T_m(i, j, P) > 0$ **do**
4:      $P \leftarrow P \cup \left\{ \arg\max_{X_i \notin P, \, X_i \in \widehat{\mathrm{Pre}}(X_j)} T_m(i, j, P) \right\}$
5: $\widehat{\mathrm{PA}}_j^h \leftarrow P$

---

[6]In practice, one could omit $\epsilon$ from the denominator as it is unusual to obtain $\min_h \mathrm{Var}_L^h = 0$ from finite samples and computational precision.

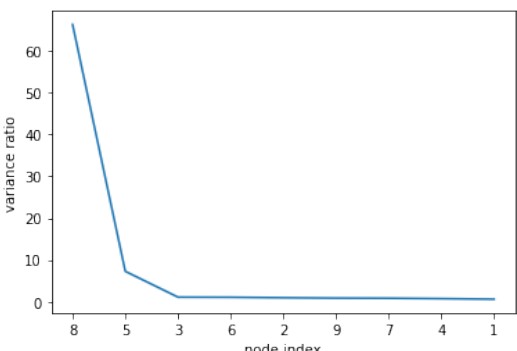

Figure 5: Statistic in eq.(8) for each node sorted in non-increasing order. In this case, node index 5 corresponds to the *elbow point*, allowing us to estimate nodes 5 and 8 as shifted nodes.

# B    Detailed Proofs

## B.1    Proof of Theorem 1

To prove Theorem 1 we will make use of the following lemmas.

**Lemma 1.** *Let $\{a_h\}_{h=1}^{H}$ and $\{b_h\}_{h=1}^{H}$ be two sequences of real numbers, where $a_h > 0, \forall h$. Then we have:*

$$\left(\sum_{h=1}^{H} a_h b_h^2\right)\left(\sum_{h=1}^{H} a_h\right) - \left(\sum_{h=1}^{H} a_h b_h\right)^2 \geq 0,$$

*with equality if and only if $b_i = b_j, \forall j \neq i \in [H]$.*

*Proof.* We can invoke the Cauchy–Schwarz inequality with vectors $\boldsymbol{u} = (\sqrt{a_1}, \ldots, \sqrt{a_H})$, and $\boldsymbol{v} = (b_1\sqrt{a_1}, \ldots, b_H\sqrt{a_H})$, then we have:

$$(\boldsymbol{u}^\top \boldsymbol{v})^2 \leq \|\boldsymbol{u}\|_2^2 \|\boldsymbol{v}\|_2^2,$$

which proves the inequality. The equality holds if and only if $\boldsymbol{u}$ and $\boldsymbol{v}$ are linearly dependent, i.e., when $b_i = b_j$ for all $i \neq j \in [H]$.    □

**Lemma 2.** *For any $j$, if $\mathbb{P}^h(X_j \mid \mathrm{PA}_j^h) = \mathbb{P}^{h'}(X_j \mid \mathrm{PA}_j^{h'})$, then $c_j^h = c_j^{h'}$.*

*Proof.* Denote the associated density of $\mathbb{P}^h(X_j \mid \mathrm{PA}_j^h)$ when $X_j = x_j$ as $p_{N_j}^h(x_j - f_j^h(\mathrm{PA}_j^h))$ and let $u = x_j - f_j^h(\mathrm{PA}_j^h)$

$$\frac{\partial^2}{\partial(x_j)^2} \log p_{N_j}^h(x_j - f_j^h(\mathrm{PA}_j^h))$$

$$= \frac{\partial \log p_{N_j}^h(u)}{\partial u} \frac{\partial^2 u}{\partial(x_j)^2} + \frac{\partial^2 \log p_{N_j}^h(u)}{\partial u^2}\left(\frac{\partial u}{\partial x_j}\right)^2$$

$$= 0 + c_j^h = c_j^h$$

where we use the fact that $\frac{\partial u}{\partial x_j} = 1, \frac{\partial^2 u}{\partial(x_j)^2} = 0$. Then it immediate follows that if $\mathbb{P}^h(X_j \mid \mathrm{PA}_j^h) = \mathbb{P}^{h'}(X_j \mid \mathrm{PA}_j^{h'})$, then $c_j^h = c_j^{h'}$    □

**Lemma 3.** *For any $j$, under Assumption B, $\frac{\partial}{\partial x_j} \log p_{N_j}^h(x_j - f_j^h(\mathrm{PA}_j^h)) = \frac{\partial}{\partial x_j} \log p_{N_j}^{h'}(x_j - f_j^{h'}(\mathrm{PA}_j^{h'}))$ if and only if $\mathbb{P}^h(X_j \mid \mathrm{PA}_j^h) = \mathbb{P}^{h'}(X_j \mid \mathrm{PA}_j^{h'})$, where $p^h$ and $p^{h'}$ are the probability density functions corresponding to the probability measures $\mathbb{P}^h$ and $\mathbb{P}^{h'}$ when $X_j = x_j$.*

*Proof.* Denote the associated density of $\mathbb{P}^h(X_j \mid \mathrm{PA}_j^h)$ when $X_j = x_j$ as $p_{N_j}^h(x_j - f_j^h(\mathrm{PA}_j^h))$, we proceed as follows:

$$\frac{\partial}{\partial x_j} \log p_{N_j}^h(x_j - f_j^h(\mathrm{PA}_j^h)) = \frac{\partial}{\partial x_j} \log p_{N_j}^{h'}(x_j - f_j^{h'}(\mathrm{PA}_j^{h'}))$$

$$\Longleftrightarrow \log p_{N_j}^h(x_j - f_j^h(\mathrm{PA}_j^h)) = \log p_{N_j}^{h'}(x_j - f_j^{h'}(\mathrm{PA}_j^{h'})) + const$$

$$\Longleftrightarrow p_{N_j}^h(x_j - f_j^h(\mathrm{PA}_j^h)) = p_{N_j}^{h'}(x_j - f_j^{h'}(\mathrm{PA}_j^{h'})) \cdot e^{const}$$

$$\Rightarrow \int_{\mathbb{R}} p_{N_j}^h(x_j - f_j^h(\mathrm{PA}_j^h)) \mathrm{d}x_j = e^{const} \cdot \int_{\mathbb{R}} p_{N_j}^{h'}(x_j - f_j^{h'}(\mathrm{PA}_j^{h'})) \mathrm{d}x_j$$

$$\Rightarrow \int_{\mathbb{R}} p_{N_j}^h(x_j - f_j^h(\mathrm{PA}_j^h)) \mathrm{d}(x_j - f_j^h(\mathrm{PA}_j^h)) = e^{const} \cdot \int_{\mathbb{R}} p_{N_j}^{h'}(x_j - f_j^{h'}(\mathrm{PA}_j^{h'})) \mathrm{d}(x_j - f_j^{h'}(\mathrm{PA}_j^{h'}))$$

$$\Rightarrow 1 = 1 \cdot e^{const}$$

$$\Rightarrow const = 0$$

Here, $const$ is a constant that is independent of $x_j$. Integrating both sides with respect to $x_j$ and using the fact that $\int p^h(x) \mathrm{d}x = 1$, we conclude that $const = 0$. Hence, we can establish the following:

$$\frac{\partial}{\partial x_j} \log p_{N_j}^h(x_j - f_j^h(\mathrm{PA}_j^h)) = \frac{\partial}{\partial x_j} \log p_{N_j}^{h'}(x_j - f_j^{h'}(\mathrm{PA}_j^{h'}))$$

$$\Longleftrightarrow p_{N_j}^h(x_j - f_j^h(\mathrm{PA}_j^h)) = p_{N_j}^{h'}(x_j - f_j^{h'}(\mathrm{PA}_j^{h'}))$$

$$\Longleftrightarrow \mathbb{P}^h(X_j \mid \mathrm{PA}_j^h) = \mathbb{P}^{h'}(X_j \mid \mathrm{PA}_j^{h'})$$

$\square$

*Proof of Theorem 1.* Let us first expand the log density of the mixture distribution:

$$\log q(x) = \log \left( \sum_{h=1}^{H} w_h p^h(x) \right)$$

Then, recall that $s(x) = \nabla \log q(x)$, the $j$-entry reads:

$$s_j(x) = \sum_{h=1}^{H} \frac{w_h p^h(x)}{\sum_{k=1}^{H} w_k p^k(x)} \left[ \frac{\partial}{\partial x_j} \log p^h(x_j \mid \mathrm{PA}_j^h) + \sum_{i \in \mathrm{CH}_j^h} \frac{\partial}{\partial x_j} \log p^h(x_i \mid \mathrm{PA}_i^h) \right]$$

$$= \sum_{h=1}^{H} \frac{w_h p^h(x)}{\sum_{k=1}^{H} w_k p^k(x)} \left[ \frac{\partial}{\partial x_j} \log p_{N_j}^h \left( x_j - f_j^h(\mathrm{PA}_j^h) \right) + \sum_{i \in \mathrm{CH}_j^h} \frac{\partial}{\partial x_j} \log p_{N_i}^h \left( x_i - f_i^h(\mathrm{PA}_i^h) \right) \right]$$

$$(9)$$

**Condition (i).** First we will prove condition (i). That is, given a leaf node $X_j$ in all DAGs $G^h$, $X_j$ is not a shifted node (i.e. an invariant node) if and only if $\mathrm{Var}(\frac{\partial s_j(x)}{\partial x_j}) = 0$.

If $x_j$ is a leaf node in all the DAGs $G^h$, then $\mathrm{CH}_j^h = \emptyset, \forall h \in [H]$, and we can write eq.(9) as:

$$s_j(x) = \sum_{h=1}^{H} \frac{w_h p^h(x)}{\sum_{k=1}^{H} w_k p^k(x)} \frac{\partial}{\partial x_j} \log p_{N_j}^h \left( x_j - f_j^h(\mathrm{PA}_j^h) \right)$$

We use $\mathrm{Den}(\frac{\partial s_j(x)}{\partial x_j})$ and $\mathrm{Num}(\frac{\partial s_j(x)}{\partial x_j})$ to denote the denominator and numerator of $\frac{\partial s_j(x)}{\partial x_j}$, respectively. Then we have:

$$\mathrm{Den}(\frac{\partial s_j(x)}{x_j}) = \left( \sum_{k=1}^{H} w_k p^k(x) \right)^2$$

$$\text{Num}(\frac{\partial s_j(x)}{x_j}) = \left[\sum_{h=1}^{H} w_h p^h(x) \frac{\partial^2}{\partial x_j^2} \log p_{N_j}^h(x_j - f_j^h(\text{PA}_j^h)) + w_h p^h(x) \left(\frac{\partial}{\partial x_j} \log p_{N_j}^h(x_j - f_j^h(\text{PA}_j^h))\right)^2\right]$$

$$\times \left[\sum_{k=1}^{H} w_k p^k(x)\right] - \left[\sum_{h=1}^{H} w_h p^h(x) \frac{\partial}{\partial x_j} \log p_{N_j}^h(x_j - f_j^h(\text{PA}_j^h))\right]^2$$

Now, dividing $\text{Num}(\frac{\partial s_j(x)}{\partial x_j})$ over $\text{Den}(\frac{\partial s_j(x)}{\partial x_j})$, we obtain:

$$\frac{\partial s_j(x)}{\partial x_j} = \frac{\text{Num}(\frac{\partial s_j(x)}{x_j})}{\text{Den}(\frac{\partial s_j(x)}{x_j})} = \sum_{h=1}^{H} \frac{w_h p^h(x)}{\sum_{k=1}^{H} w_k p^k(x)} \frac{\partial^2}{\partial x_j^2} \log p_{N_j}^h(x_j - f_j^h(\text{PA}_j^h))$$

$$+ \sum_{h=1}^{H} \frac{w_h p^h(x)}{\sum_{k=1}^{H} w_k p^k(x)} \left(\frac{\partial}{\partial x_j} \log p_{N_j}^h(x_j - f_j^h(\text{PA}_j^h))\right)^2$$

$$- \left[\sum_{h=1}^{H} \frac{w_h p^h(x)}{\sum_{k=1}^{H} w_k p^k(x)} \frac{\partial}{\partial x_j} \log p_{N_j}^h(x_j - f_j^h(\text{PA}_j^h))\right]^2 \quad (10)$$

Note that since $x_j \notin \text{PA}_j^h$, the function $f_j^h(\text{PA}_j^h)$ is independent of $x_j$.

Let $a_h = w_h p^h(x)$, and let $b_h = \frac{\partial}{\partial x_j} \log p_{N_j}^h(x_j - f_j^h(\text{PA}_j^h))$. Then, the last two summands of the RHS of eq.(10) can be written as:

$$\frac{1}{\left(\sum_{h=1}^{H} a_h\right)^2} \left[\left(\sum_{h=1}^{H} a_h b_h^2\right)\left(\sum_{h=1}^{H} a_h\right) - \left(\sum_{h=1}^{H} a_h b_h\right)^2\right] \geq 0, \quad (11)$$

where the last inequality holds from Lemma 1. Then, by Lemma 3, we have that $b_h = b_{h'} \iff \mathbb{P}^h(X_j \mid \text{PA}_j^h) = \mathbb{P}^{h'}(X_j \mid \text{PA}_j^{h'})$ for all $h, h' \in [H]$. Then if $b_h = b_{h'}$ holds, by Lemma 2, we have $c_j^h = c_j^{h'} := c_j$ and then the first term for eq.(10) boils down to a constant $c_j$. Finally, from Lemma 1, we have that equality in eq.(11) holds if and only if $b_h = b_{h'}$ for all $h, h' \in [H]$. Thus, we conclude that:

If $X_j$ is a leaf node for all $G^h$, then $X_j$ is not a shifted node $\iff \frac{\partial s_j(x)}{\partial x_j} = c_j$,

where $\frac{\partial s_j(x)}{\partial x_j} = c_j$ is equivalent to $\text{Var}_q(\frac{\partial s_j(x)}{\partial x_j}) = 0$.

**Condition (ii).** We now prove that if $\text{Var}_q(\frac{\partial s_j(x)}{\partial x_j}) > 0$, then only one of the following two cases holds: Case 1) $X_j$ is a leaf node for all $G^h$ and a shifted node. Case 2) $X_j$ is not a leaf node in at least one DAG $G^h$.

Case 1 follows immediately from the proof of condition (i) above.

For Case 2, we study whether there exists a non-leaf node $X_j$ with $\text{Var}_q(\frac{\partial s_j(x)}{\partial x_j}) = 0$. Taking the partial derivative of $s_j(x)$ in eq.(9) w.r.t. $x_j$, we have:

$$\frac{\partial s_j(x)}{\partial x_j} = \sum_{h=1}^{H} \frac{w_h p^h(x)}{\sum_{k=1}^{H} w_k p^k(x)} \left(\frac{\partial^2}{\partial x_j^2} \log p_{N_j}^h(x_j - f_j^h(\text{PA}_j^h)) + \sum_{i \in \text{CH}_j^h} \frac{\partial^2}{\partial x_j^2} \log p_{N_i}^h(x_i - f_i^h(\text{PA}_i^h))\right)$$

$$+ \sum_{h=1}^{H} \frac{w_h p^h(x)}{\sum_{k=1}^{H} w_k p^k(x)} \left(\frac{\partial}{\partial x_j} \log p_{N_j}^h(x_j - f_j^h(\text{PA}_j^h)) + \sum_{i \in \text{CH}_j^h} \frac{\partial}{\partial x_j} \log p_{N_i}^h(x_i - f_i^h(\text{PA}_i^h))\right)^2$$

$$
- \left[ \sum_{h=1}^{H} \frac{w_h p^h(x)}{\sum_{k=1}^{H} w_k p^k(x)} \left( \frac{\partial}{\partial x_j} \log p_{N_j}^h(x_j - f_j^h(\mathrm{PA}_j^h)) + \sum_{i \in \mathrm{CH}_j^h} \frac{\partial}{\partial x_j} \log p_{N_i}^h(x_i - f_i^h(\mathrm{PA}_i^h)) \right) \right]^2
$$

By Assumptions B, we have $\frac{\partial^2}{\partial x_j^2} \log p_{N_j}^h(x_j - f_j^h(\mathrm{PA}_j^h)) = c_j^h$. For simplicity, let $a_h = \frac{\partial}{\partial x_j} \log p_{N_j}^h(x_j - f_j^h(\mathrm{PA}_j^h)) + \sum_{i \in \mathrm{CH}_j^h} \frac{\partial}{\partial x_j} \log p_{N_i}^h(x_i - f_i^h(\mathrm{PA}_i^h))$. Then, we have:

$$
\frac{\partial s_j(x)}{\partial x_j} = \sum_{h=1}^{H} \frac{w_h p^h(x)}{\sum_{k=1}^{H} w_k p^k(x)} c_j^h + \underbrace{\sum_{h=1}^{H} \frac{w_h p^h(x)}{\sum_{k=1}^{H} w_k p^k(x)} \sum_{i \in \mathrm{CH}_j^h} \frac{\partial^2}{\partial x_j^2} \log p_{N_i}^h(x_i - f_i^h(\mathrm{PA}_i^h))}_{\text{term 1}}
$$

$$
+ \underbrace{\sum_{h=1}^{H} \frac{w_h p^h(x)}{\sum_{k=1}^{H} w_k p^k(x)} a_h^2 - \left( \sum_{h=1}^{H} \frac{w_h p^h(x)}{\sum_{k=1}^{H} w_k p^k(x)} a_h \right)^2}_{\text{term 2}}. \quad (12)
$$

We prove that $\frac{\partial^2}{\partial x_j^2} \log p_{N_i}^h(x_i - f_i^h(\mathrm{PA}_i^h))$, is not constant under any circumstance, by contradiction. Let $G^h$ be an environment's DAG where $X_j$ is not a leaf, and let $X_u \in \mathrm{CH}_j^h$ such that $X_u \notin \cup_{i \in \mathrm{CH}_j^h} \mathrm{PA}_i^h$. Note that $X_u$ always exist since $X_j$ is not a leaf, and it suffices to pick a child $X_u$ appearing at the latest position in the topological order of $G^h$. Now suppose that $\frac{\partial^2}{\partial x_j^2} \log p_{N_u}^h(x_u - f_u^h(\mathrm{PA}_u^h)) = a$, where $a$ is a constant. Then we have:

$$
\frac{\partial}{\partial x_j} \log p_{N_u}^h(x_u - f_u^h(\mathrm{PA}_u^h)) = a x_j + g(x_{-j}),
$$

$$
\frac{\partial}{\partial x_j} f_u^h(\mathrm{PA}_u^h) \cdot \frac{\partial}{\partial n_u} \log p_{N_u}^h(n_u) = a x_j + g(x_{-j}).
$$

By deriving both sides w.r.t. $x_u$, we obtain:

$$
\frac{\partial}{\partial x_j} f_u^h(\mathrm{PA}_u^h) \cdot \frac{\partial^2}{\partial n_u^2} \log p_{N_u}^h(n_u) = \frac{\partial g(x_{-j})}{\partial x_u}
$$

$$
\frac{\partial}{\partial x_j} f_u^h(\mathrm{PA}_u^h) \cdot c_j^h = \frac{\partial g(x_{-j})}{\partial x_u}.
$$

Since the RHS does not depend on $x_j$, then $\frac{\partial f_u^h}{\partial x_j}$ cannot depend on $x_j$ neither, implying that $f_u^h$ is linear in $x_j$, thus contradicting the non-linearity assumption (Assumption A). Consequently, it becomes evident that term 1 cannot be a constant, regardless of whether the node $X_j$ has undergone a shift or not.

Now let us take a look to term 2 in eq.(12). We have:

$$
\sum_{h=1}^{H} \frac{w_h p^h(x)}{\sum_{k=1}^{H} w_k p^k(x)} a_h^2 - \left( \sum_{h=1}^{H} \frac{w_h p^h(x)}{\sum_{k=1}^{H} w_k p^k(x)} a_h \right)^2 \geq 0,
$$

where the inequality follows by Jensen's inequality. Thus we conclude that if $X_j$ is a non-leaf node, we have $\mathrm{Var}_q(\frac{\partial s_j(x)}{\partial x_j}) > 0$. $\qquad \square$

## B.2 Proof of Theorem 2

To proof the theorem we will need the following assumptions:

**Assumption C.** *Let* $\mathrm{MB}_j^h$ *denote the Markov Blanket of node* $X_j$ *under environment h, then assume*

- *There are non-negative real number $\beta$ and $C$ such that for any subset $X_{\boldsymbol{S}} \subseteq \mathrm{Pre}(X_j^h)$ of size $\leq 1/\delta + 2$, any $x, x' \in \mathbb{R}^{|X_{\boldsymbol{S}}|}$ and any $t \in \mathbb{R}$,*

$$| \, \mathbb{P}(X_j^h \geq t \mid X_{\boldsymbol{S}} = x) - \mathbb{P}(X_j^h \geq t \mid X_{\boldsymbol{S}} = x') \, | \leq C(1 + \|x\|^{\beta} + \|x'\|^{\beta})\|x - x'\|$$

  *where $|X_{\boldsymbol{S}}|$ is the size of the set $X_{\boldsymbol{S}}$.*

- *There are positive numbers $C_1$ and $C_2$ such that for any $X_{\boldsymbol{S}}$ of size $\leq 1/\delta + 2$ and any $t > 0$, $\mathbb{P}(\|X_{\boldsymbol{S}}\| \geq t) \geq C_1 e^{-C_2 t}$*

- *For any subset $X_{\boldsymbol{S}} \subseteq \mathrm{Pre}(X_j^h)$ such that $X_{\boldsymbol{S}} \subsetneq \mathrm{MB}_j^h$, there exists $X_i$ with $X_i \in \mathrm{MB}_j^h \backslash X_{\boldsymbol{S}}$, such that for any $X_j$ with $X_j \notin \mathrm{MB}_j^h$,*

$$Q(X_{\boldsymbol{S}} \cup \{X_i\}) - Q(X_{\boldsymbol{S}} \cup \{X_j\}) \geq \delta/4 \qquad Q(X_{\boldsymbol{S}}) = \int \mathrm{Var}(\mathbb{P}(X_j^h \geq t \mid X_S)) d\mu(t)$$

*where $\delta$ is the largest number such that for any subset $X_{\boldsymbol{S}}$ from $\mathrm{Pre}(X_j^h)$, there is some $X_i \notin X_{\boldsymbol{S}}$ such that $Q(X_{\boldsymbol{S}} \cup \{X_i\}) \geq Q(X_{\boldsymbol{S}}) + \delta$.*

*Proof.* Under Assumption C, from Theorem 3.1 in [4] , we have

$$\mathbb{P}(\widehat{\mathrm{MB}}_j^h = \mathrm{MB}_j^h) \geq 1 - C_3 e^{-C_4 m}$$

where $C_3$ and $C_4$ are constants that depend only on the data generation process, and $m$ is the number of samples. Since the estimated topological order $\hat{\pi}$ is assumed to be valid for all environments, we can conclude that node $X_j$ is a leaf node in the input data $\{\mathrm{Pre}(X_j^h), X_j^h\}$ for all $h$. As a result, we have $\mathrm{MB}_j^h = \mathrm{PA}_j^h$ based on the Markov blanket definition. Therefore, the output of Algorithm 4 is equal to the true parent set $\mathrm{PA}_j^h$ with high probability. $\qquad\square$

## B.3   Proof of Theorem 3 in Appendix D

To prove Theorem 3 we will make use of the following lemmas.

**Lemma 4.** *Suppose $\boldsymbol{X}$ is an $n \times k_x$ dimension matrix, $\boldsymbol{Y}$ is an $n \times k_y$ dimension matrix. Let the columns of the concatenated matrix $\boldsymbol{Z} = (\boldsymbol{X}, \boldsymbol{Y})$ be linearly independent. Consider $\tilde{\beta}_x$, a $k_x$-dimensional vector, and $\tilde{\beta}_y$ and $\beta_y$, both $k_y$-dimensional vectors. If $\boldsymbol{X}\tilde{\beta}_x + \boldsymbol{Y}\tilde{\beta}_y = \boldsymbol{Y}\beta_y$, then it follows that $\tilde{\beta}_y = \beta_y$ and $\tilde{\beta}_x = \boldsymbol{0}$.*

*Proof.*

$$\boldsymbol{X}\tilde{\beta}_x + \boldsymbol{Y}\tilde{\beta}_y - \boldsymbol{Y}\beta_y = (\boldsymbol{X}, \boldsymbol{Y})\begin{pmatrix} \tilde{\beta}_x \\ \tilde{\beta}_y \end{pmatrix} - (\boldsymbol{X}, \boldsymbol{Y})\begin{pmatrix} \boldsymbol{0}_{k_x} \\ \beta_y \end{pmatrix} = \boldsymbol{Z}\begin{pmatrix} \tilde{\beta}_x \\ \tilde{\beta}_y - \beta_y \end{pmatrix} = 0$$

Since $\boldsymbol{Z}$ has full column rank, then the null space of $\boldsymbol{Z}$ is $\boldsymbol{0}$, which implies $\tilde{\beta}_x = \boldsymbol{0}$, $\tilde{\beta}_y = \beta_y$. $\qquad\square$

**Lemma 5.** *For any $h \in [H]$, if*

$$\sum_{k \in \mathrm{Pre}(X_j)} \Psi_{jk}\tilde{\beta}_{jk}^h = \sum_{k \in \mathrm{PA}_j^h} \Psi_{jk}\beta_{jk}^h,$$

*then $\tilde{\beta}_{jk}^h = \beta_{jk}^h$ if $k \in \mathrm{PA}_j^h$, and $\tilde{\beta}_{jk}^h = 0$ if $k \notin \mathrm{PA}_j^h$.*

*Proof.* Rearrange the set $\mathrm{Pre}(X_j)$ so that $\mathrm{Pre}(X_j) = \{X_{k_1}, X_{k_2}, \ldots, X_{k_m}, X_{k_{m+1}}, \ldots, X_{k_p}\}$, where $\{X_{k_1}, \ldots, X_{k_m}\} = \mathrm{Pre}(X_j) \setminus \mathrm{PA}_j$, and $\{X_{k_{m+1}}, \ldots, X_{k_p}\} = \mathrm{PA}_j$. Then let $\boldsymbol{X} = (\Psi_{k_1}, \ldots, \Psi_{k_m})$, $\boldsymbol{Y} = (\Psi_{k_{m+1}}, \ldots, \Psi_{k_p})$, and $\boldsymbol{Z} = (\boldsymbol{X}, \boldsymbol{Y})$. By the linear independence property of the basis functions of $\mathrm{Pre}(X_j)$, we have that the columns of $\boldsymbol{Z}$ are linearly independent. Also, let

$$\tilde{\beta}_x = \begin{pmatrix} \tilde{\beta}_{jk_1}^h \\ \vdots \\ \tilde{\beta}_{jk_m}^h \end{pmatrix}, \quad \tilde{\beta}_y = \begin{pmatrix} \tilde{\beta}_{jk_{m+1}}^h \\ \vdots \\ \tilde{\beta}_{jk_p}^h \end{pmatrix}, \quad \beta_y = \begin{pmatrix} \beta_{jk_{m+1}}^h \\ \vdots \\ \beta_{jk_p}^h \end{pmatrix}$$

Then we must have:

$$\sum_{k \in \mathrm{Pre}(X_j)} \Psi_{jk} \tilde{\beta}_{jk}^h = \sum_{k \in \mathrm{PA}_j^h} \Psi_{jk} \beta_{jk}^h \quad \Rightarrow \quad \boldsymbol{X} \tilde{\beta}_x + \boldsymbol{Y} \tilde{\beta}_y = \boldsymbol{Y} \beta_y.$$

Then by Lemma 4, we have $\tilde{\beta}_{jk}^h = \beta_{jk}^h$ if $k \in \mathrm{PA}_j^h$, $\tilde{\beta}_{jk}^h = 0$ if $k \notin \mathrm{PA}_j^h$. $\qquad\square$

*Proof of Theorem 3.* We know that $\mathrm{Pre}(X_j)$ contains the ancestors of $X_j$. Then, in environment $h$, we have:

$$\mathbb{E}_{p^h}[X_j \mid \mathrm{Pre}(X_j)] = \mathbb{E}_{p^h}[f_j^h(\mathrm{PA}_j^h) \mid \mathrm{Pre}(X_j)] + \mathbb{E}_{p^h}[N_j \mid \mathrm{Pre}(X_j)]$$
$$= f_j^h(\mathrm{PA}_j^h),$$

where the last equality follows since the first conditional expectation is equal to $f_j^h(\mathrm{PA}_j)$, due to $\mathrm{PA}_j \subseteq \mathrm{Pre}(X_j)$. Moreover, in the second conditional expectation term, we have that $N_j$ is marginally independent of $\mathrm{Pre}(X_j)$ by the d-separation criterion. Thus the conditional expectation of $N_j$ equals to the marginal expectation of $N_j$, which is 0. Finally,

$$\sum_{k \in \mathrm{Pre}(X_j)} \Psi_{jk} \tilde{\beta}_{jk}^h = \mathbb{E}_{p^h}[X_j \mid \mathrm{Pre}(X_j)] = f_j^h(\mathrm{PA}_j^h) = \sum_{k \in \mathrm{PA}_j^h} \Psi_{jk} \beta_{jk}^h$$

By Lemma 5, we have $\tilde{\beta}_{jk}^h = \beta_{jk}^h$ if $k \in \mathrm{PA}_j^h$, $\tilde{\beta}_{jk}^h = 0$ if $k \notin \mathrm{PA}_j^h$. $\qquad\square$

## C  Additional Experiments

This section provides a thorough evaluation of the pipeline of our method. We begin by assessing the performance of our method in detecting the shifted nodes. Subsequently, we extend the evaluation to include the recovery of the structurally shifted edges.

### C.1  Experiments on detecting shifted nodes

**Graph models.** We ran experiments by generating adjacency matrices using the Erdős–Rényi (ER) and Scale free (SF) graph models. For a given number of variables $d$, ER$k$ and SF$k$ indicate an average number of edges equal to $kd$.

**Data generation process.** We first sampled a Directed Acyclic Graph (DAG) according to either the Erdős-Rényi (ER) model or the Scale-Free (SF) model for environment $\mathcal{E}_1$.

For environment $\mathcal{E}_2$, we used the same DAG structure as in environment $\mathcal{E}_1$, ensuring a direct comparison between the two environments. To introduce artificial shifted nodes, we randomly selected $0.2 \cdot d$ nodes from the non-root nodes, where $d$ represents the total number of nodes in the DAG. These selected nodes were considered as the "shifted nodes," denoted as $S$, with $|S| = 0.2d$.

The functional relationship between a node $X_j$ and its parents in environment $\mathcal{E}_1$ was defined as follows:

$$X_j = \sum_{i \in \mathrm{PA}_j} \sin(X_i^2) + N_j,$$

while for environment $\mathcal{E}_2$, we defined the functional relationships between each node and its parents by:

$$X_j = \begin{cases} \sum_{i \in \mathrm{PA}_j} \sin(X_i^2) + N_j, & \text{if} \quad X_j \notin S, \\ \sum_{i \in \mathrm{PA}_j} 4\cos(2X_i^2 - 3X_i) + N_j, & \text{if} \quad X_j \in S. \end{cases}$$

**Experiment detail.** In each simulation, we generated 500 data points, with the variances of the noise set to 1. We conducted 30 simulations for each combination of graph type, noise type, and number of nodes. The running time was recorded by executing the experiments on an Intel Xeon Gold 6248R Processor with 8 cores. For our method, we used the hyperparameters `eta_G` = 0.005, `eta_H` = 0.005, and threshold $t = 2$ (see Algorithm 3).

**Evaluation.** We conducted a comparative analysis to evaluate the performance of our method in detecting shifted nodes compared to DCI. The evaluation was based on F1 score, precision, and recall as the evaluation metrics. Furthermore, we examined the robustness of our method by conducting tests using Gumbel and Laplace as noise distributions.

Figures 6, 7, and 8 illustrate our method's performance across varying numbers of nodes and sparsity levels of the graphs. Our method consistently outperformed DCI in terms of F1 score, precision, and recall.

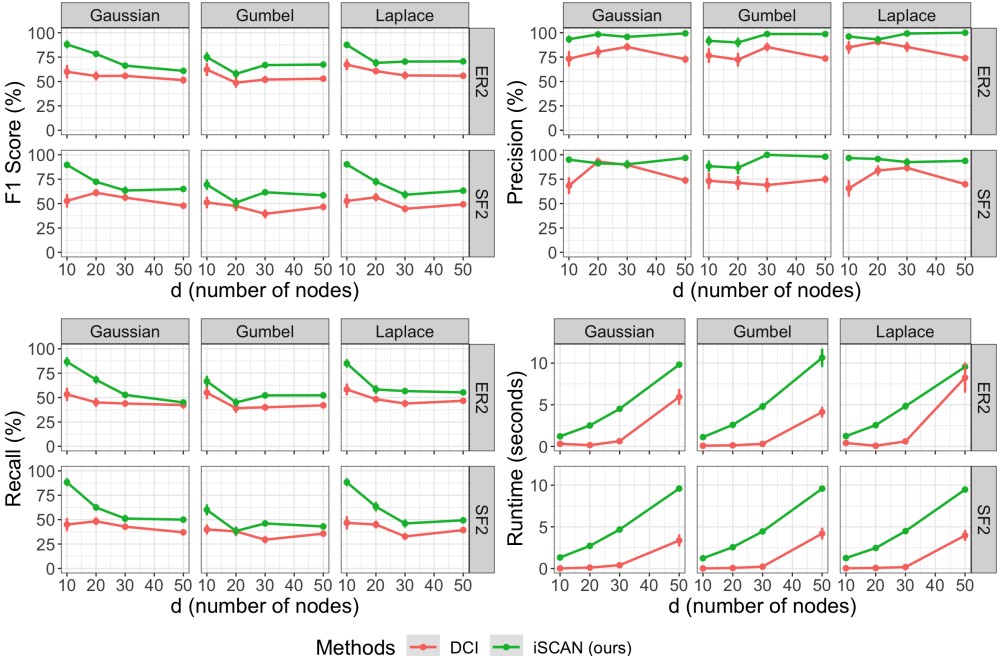

Figure 6: Shifted nodes detection in ER2 and SF2 graphs. For each point, we conducted 30 simulations as described in Section C.1. The points indicate the average values obtained from these simulations, while the error bars depict the standard errors. For each simulation, 500 samples were generated. Our method iSCAN (green) consistently outperformed DCI (red) in terms of F1 score, precision, and recall.

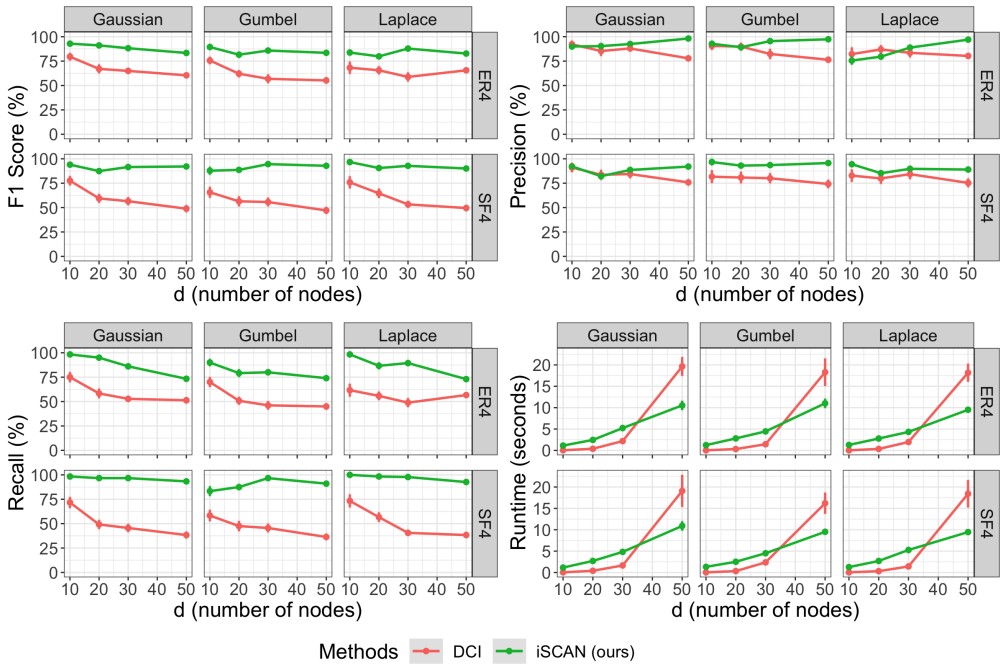

Figure 7: Shifted nodes detection in ER4 and SF4 graphs. For each point, we conducted 30 simulations as described in Section C.1. The points indicate the average values obtained from these simulations, while the error bars depict the standard errors. For each simulation, 500 samples were generated. Our method iSCAN (green) consistently outperformed DCI (red) in terms of F1 score, precision, and recall.

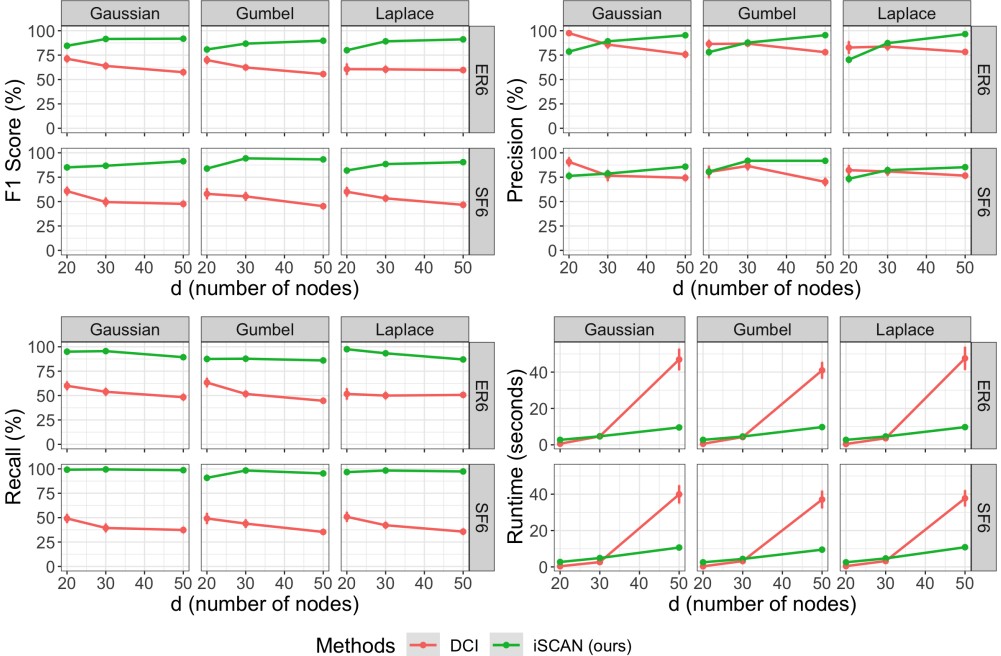

Figure 8: Shifted nodes detection in ER6 and SF6 graphs. For each point, we conducted 30 simulations as described in Section C.1. The points indicate the average values obtained from these simulations, while the error bars depict the standard errors. For each simulation, 500 samples were generated. Our method iSCAN (green) consistently outperformed DCI (red) in terms of F1 score, precision, and recall.

### C.1.1 Experiments in detecting shifted nodes from Gaussian process

**Data generation process.** We first sampled a DAG according to the ER or SF model. In our experiment, we considered two environments, $\mathcal{E}_1$ and $\mathcal{E}_2$, with the same DAG structure. Each node in the graph had a functional relationship with its parents defined as $X_j = f_j^h(\text{PA}_j^h) + N_j$, where $N_j$ is an independent standard Gaussian variable. Recall that the superscript $h$ denotes the function for environment $\mathcal{E}_h$.

To introduce shifted nodes, we randomly selected $0.2 \cdot d$ nodes from the non-root nodes, denoted as $S$, to be the shifted nodes. In other words, $|S| = 0.2d$. For the non-shifted nodes $X_j$ (i.e $j \notin S$), we set $f_j^1 = f_j^2$. However, for each shifted node $X_j$ in $S$, we changed its functional relationship with its parents to $X_j = 2 \cdot f_j^2(\text{PA}_j^2) + N_j$.

To test our method in a more general setting involving nonlinear functions, we followed the approach in [65, 39]. Specifically, for *non-shifted nodes*, we generated the link functions $f_j^1$ by sampling Gaussian processes with a half unit bandwidth RBF kernel, and we set $f_j^2 = f_j^1$. *For shifted nodes*, $X_j \in S$, we generated the link functions $f_j^1$ and $f_j^2$ by sampling Gaussian processes with a half unit bandwidth RBF kernel independently. This allowed us to simulate different functional relationships for the shifted nodes across the two environments.

**Experiment detail.** In each simulation, we generated 1000 data points, with the variances of the noise set to 1. We conducted 30 simulations for each combination of graph type, noise type, and number of nodes. The running time was recorded by executing the experiments on an Intel Xeon Gold 6248R Processor with 8 cores. For our method, we used the hyperparameters `eta_G` $= 0.005$, `eta_H` $= 0.005$, and `elbow` $=$ True (see Remark 6).

**Evaluation.** We conducted a comparative performance analysis between our proposed Algorithm 1 (iSCAN, green) and the DCI (red) method. The results for ER2 and SF2 graphs under Gaussian, Gumbel, and Laplace noise distributions are shown in Figure 9. In certain cases, our method may underperform DCI in terms of precision, resulting in a lower F1 score. However, it is important to note that our method consistently outperforms DCI in terms of recall score.

Furthermore, Figure 10 and Figure 11 present the results for ER4/SF4, and ER6/SF6 graphs. In terms of precision, our method exhibits competitive performance and, in many cases, outperforms DCI. Notably, iSCAN consistently surpasses DCI in terms of recall score and F1 score.

These findings emphasize the strengths of our proposed method in accurately detecting shifted nodes and edges, particularly in terms of recall and overall performance. In denser graphs, our method demonstrates a superior ability to recover shifted nodes compared to DCI. This suggests that our method is well-suited for scenarios where the graph structure is more complex and contains a larger number of nodes and edges. The improved performance of our method in such settings further highlights its potential in practical applications and its ability to handle more challenging tasks.

**Top-k precision.** We have observed that in some cases, the precision of our method underperformed DCI. We attribute this to the elbow method rather than $\text{stats}_L$. To further investigate this, we conducted an analysis using only $\text{stats}_L$ and measured the precision based on different criteria. Specifically, we identified nodes as shifted if their $\text{stats}_L$ ranked first, first two, or within the top k, denoted as top-1 precision, top-2 precision, and top-k precision, respectively, where $k = |S|$.

Figure 12 presents the results of precision for top-1, top-2, and top-k criteria under various graph models and noise combinations. In most cases, the precision exceeds $80\%$ and even approaches $100\%$. These results indicate that when using $\text{stats}_L$ alone, our method still provides accurate information about shifted nodes. The findings suggest that the lower precision observed in Figure 9 can be attributed to the elbow strategy rather than the effectiveness of $\text{stats}_L$. Overall, this analysis strengthens the reliability and usefulness of $\text{stats}_L$ in accurately identifying shifted nodes in our method.

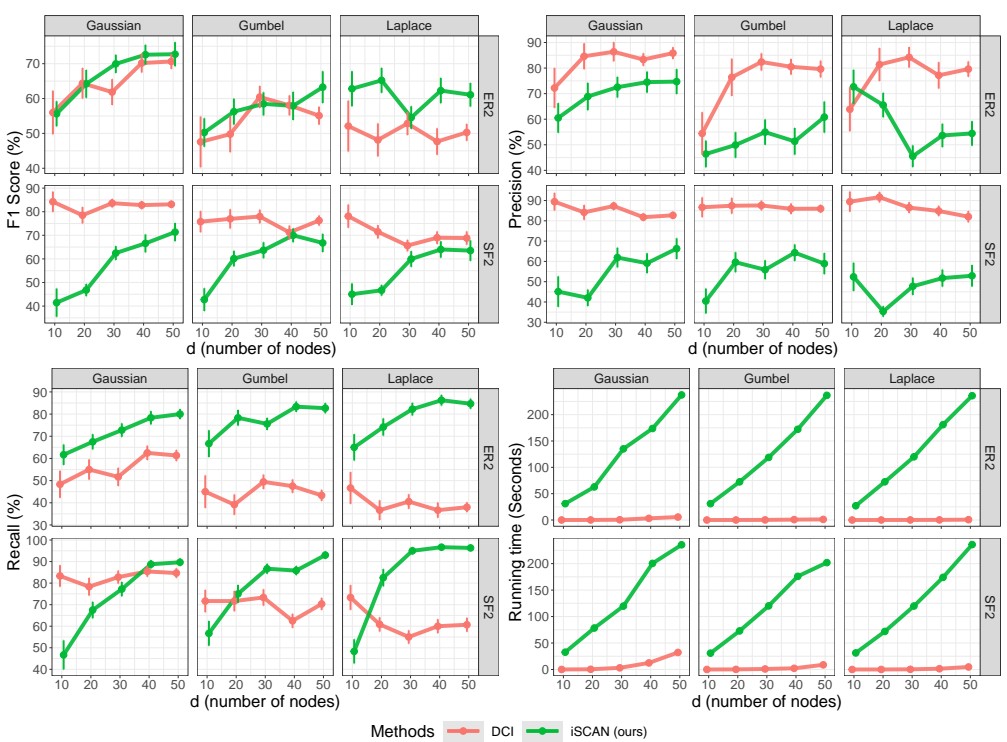

Figure 9: Experiments on detection of shifted nodes in ER2/SF2 graphs using Gaussian processes. Details described in Appendix C.1.1. The error bars represent the standard errors.

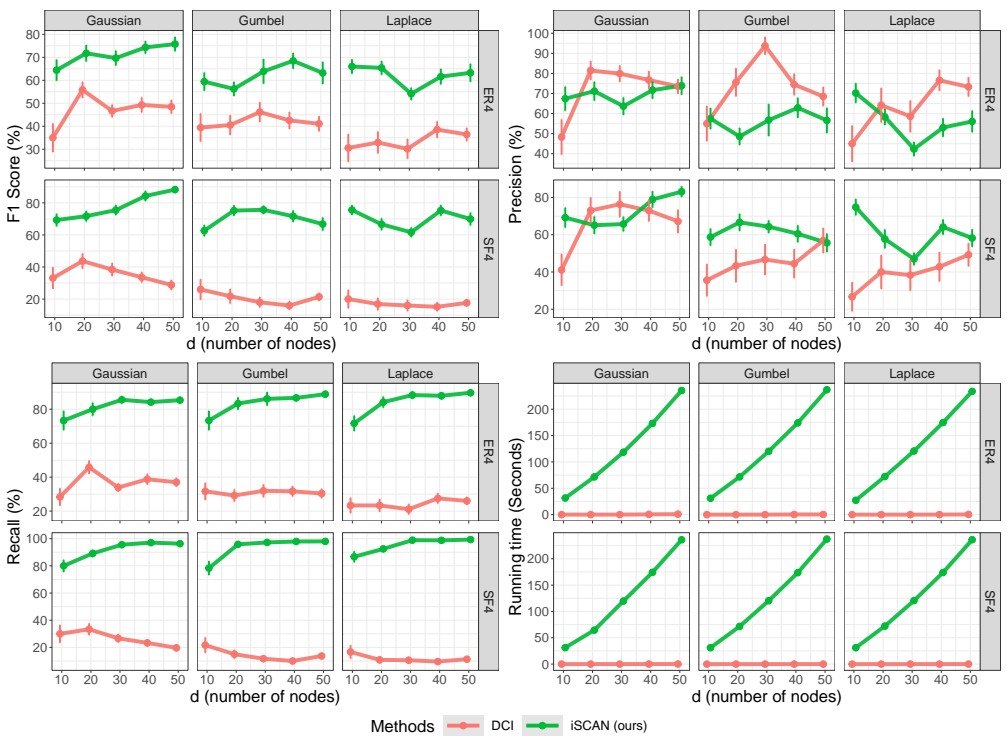

Figure 10: Experiments on detection of shifted nodes in ER4/SF4 graphs using Gaussian processes. Details described in Appendix C.1.1. The error bars represent the standard errors.

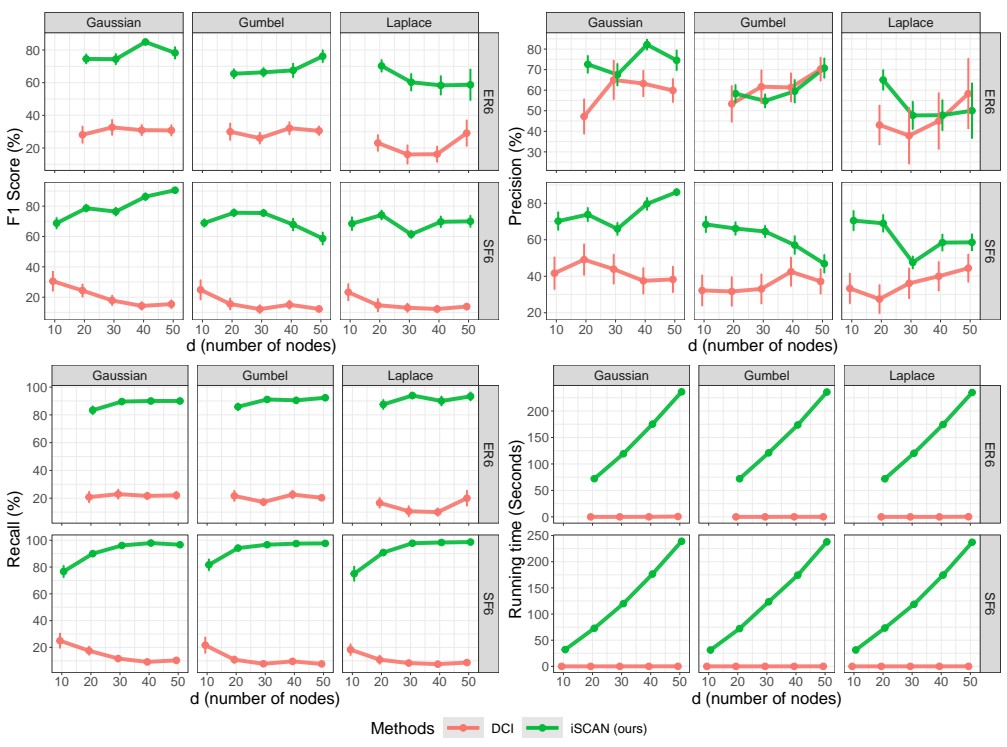

Figure 11: Experiments on detection of shifted nodes in ER6/SF6 graphs using Gaussian processes. Details described in Appendix C.1.1. The error bars represent the standard errors.

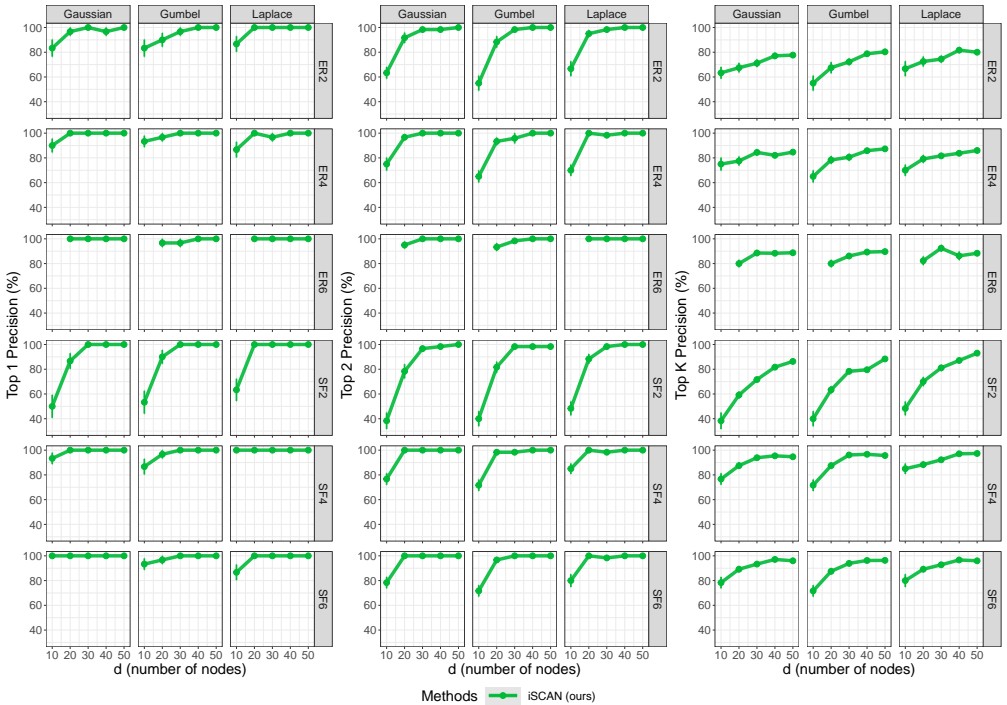

Figure 12: Top 1, 2 and K performance of iSCAN where functionals are sampled from Gaussian processes. Details described in Appendix C.1.1. The error bars represent the standard errors.

## C.2 Experiments on estimating structural shifts

**Data generation.** We first sampled a DAG, $G^1$, of $d$ nodes according to either the ER or SF model for env. $\mathcal{E}_1$. For env. $\mathcal{E}_2$, we initialized its DAG structure from env. $\mathcal{E}_1$ and produced structural changes by randomly selecting $0.2 \cdot d$ nodes from the non-root nodes. This set of selected nodes $S$, with cardinality $|S| = 0.2d$, correspond to the set of "shifted nodes". In env. $\mathcal{E}_2$, for each shifted node $X_j \in S$, we uniformly at random deleted at most three of its incoming edges, and use $D_j$ to denote the parents whose edges to $X_j$ were deleted; thus, the DAG $G^2$ is a subgraph of $G^1$. Then, in $\mathcal{E}_1$, each $X_j$ was defined as follows:

$$X_j = \sum_{i \in \mathrm{PA}_j^1 \setminus D_j} \sin(X_i^2) + \sum_{i \in D_j} 4\cos(2X_i^2 - 3X_i) + N_j$$

In $\mathcal{E}_2$, each $X_j$ was defined as follows:

$$X_j = \sum_{i \in \mathrm{PA}_j^2} \sin(X_i^2) + N_j$$

**Experiment details.** For each simulation, we generated 500 data points per environment, i.e., $m_1 = 500, m_2 = 500$ and $m = 1000$. The noise variances were set to 1. We conducted 30 simulations for each combination of graph type (ER or SF), noise type (Gaussian, Gumbel, and Laplace), and number of nodes ($d \in \{10, 20, 30, 50\}$). The running time was recorded by executing the experiments on an Intel Xeon Gold 6248R Processor with 8 cores. For our method, we used $\eta = 0.05$ for eq.(6) and eq.(7), and a threshold $t = 2$ (see Alg. 3).

In the case of the method introduced by Budhathoki et al. [11], we employed Kernel Conditional Independence (KCI) tests [86] for conducting conditional independence tests. As for CITE, KCD, UT-IGSP, and SCORE, we used their respective default parameter settings provided within their packages. Additionally, for SCORE, we employed it to estimate the DAGs independently for different environments and then compared the recovered DAGs to identify the shifted nodes. Given that Budhathoki's and KCD methods require information about the parents $\mathrm{PA}_j$ for each node $X_j$, we employed the SCORE method to find the parent sets $\mathrm{PA}_j$.

**Evaluation.** In this experiment, we assessed the performance of our method in two aspects: detecting shifted nodes and recovering the structural changes (**difference DAG**). For the evaluation of shifted node detection, we measured F1 score, recall, and precision. In the evaluation of difference DAG recovery, we compared the estimated difference DAG with the ground truth difference DAG using F1 score. Additionally, we considered the running time of the methods as another evaluation criterion.

Figures 13, 14, and 15 illustrate our method's performance in detecting shifted nodes across varying numbers of nodes and sparsity levels of the graphs. Our method consistently outperformed baselines in terms of F1 score, precision, and recall.

Figures 16 showcase the performance of our method in recovering difference DAG across different noise distribution, different numbers of nodes and sparsity levels in the graphs. Our method achieves higher F1 score in recovering the difference DAG compared with DCI.

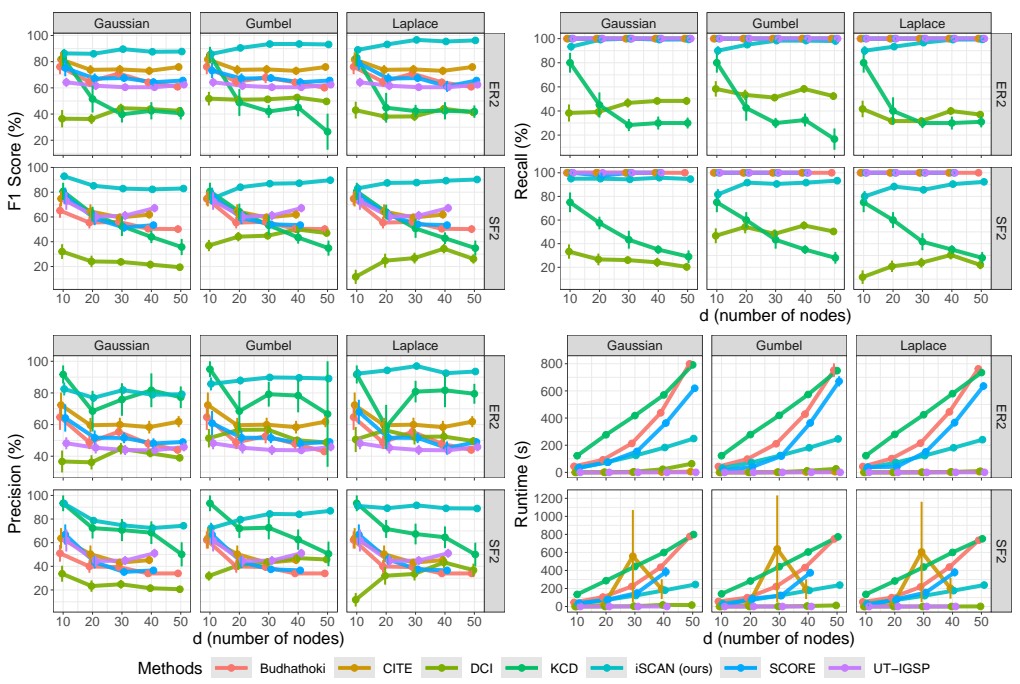

Figure 13: Shifted nodes detection performance in ER2/SF2. See App. C.2 for experimental details. iSCAN (light blue) consistently outperformed baselines in terms of F1 score, precision, and recall.

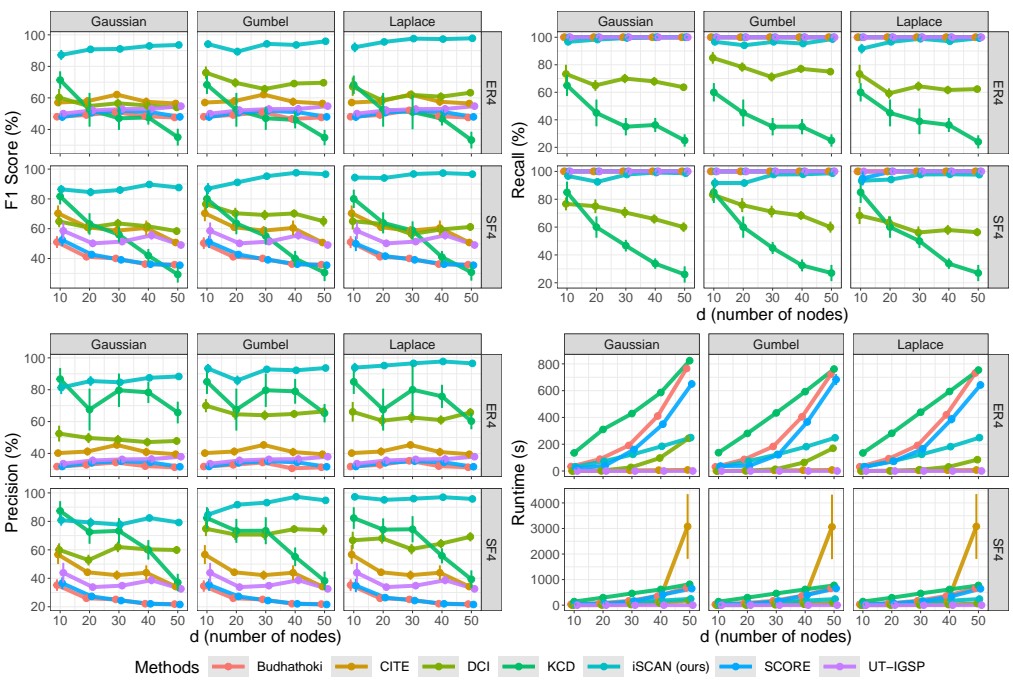

Figure 14: Shifted nodes detection performance in ER4/SF4. See App. C.2 for experimental details. iSCAN (light blue) consistently outperformed baselines in terms of F1 score, precision, and recall.

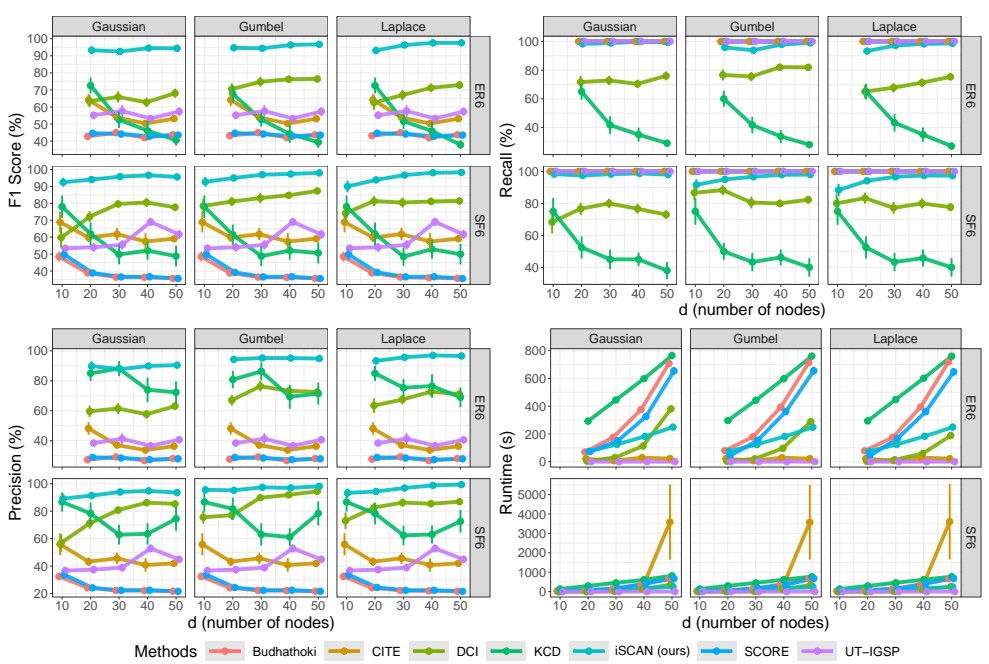

Figure 15: Shifted nodes detection performance in ER6/SF6. See App. C.2 for experimental details. iSCAN consistently outperformed baselines in terms of F1 score, precision, and recall.

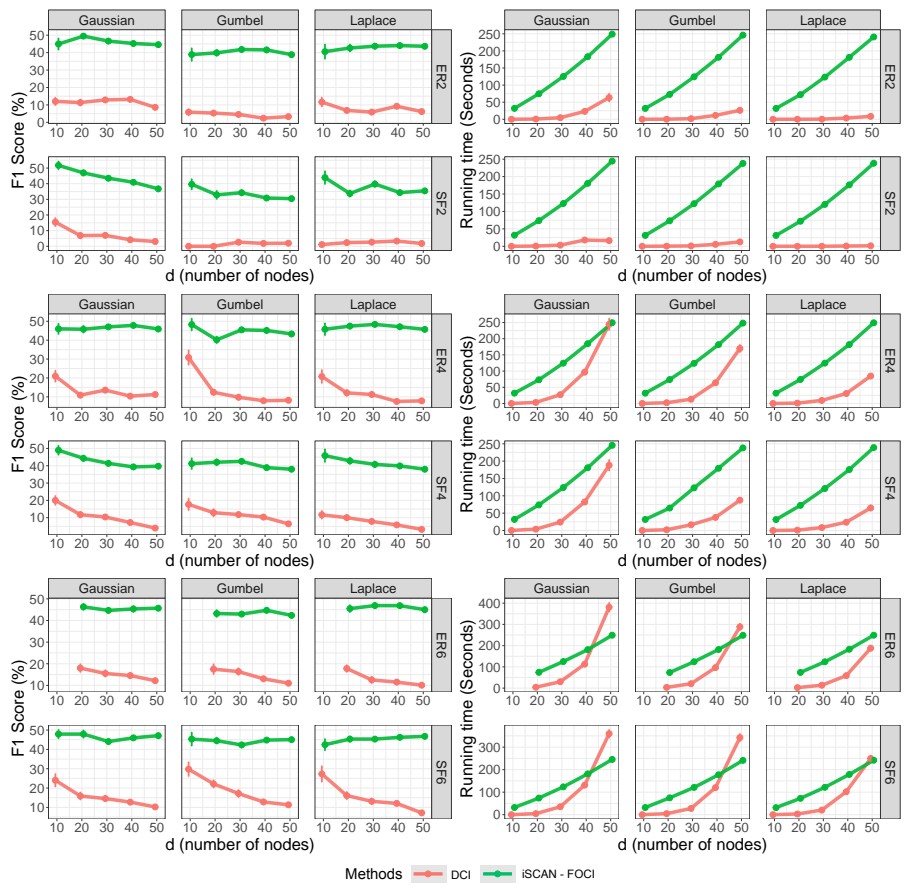

Figure 16: Difference DAG recovery performance in all different graphs. iSCAN-FOCI (green) consistently outperformed DCI (red) in terms of F1 score.

## C.3 Performance of Alg. 3 using the elbow method

In this section, we aim to understand the performance of our method when using the elbow approach discussed in Remark 6, random functions for shifted nodes, and different noise variances per variable within an environment.

**Data generation process.** We first sampled a Directed Acyclic Graph (DAG) according to either the Erdős-Rényi (ER) model or the Scale-Free (SF) model for environment $\mathcal{E}_1$.

For environment $\mathcal{E}_2$, we used the same DAG structure as in environment $\mathcal{E}_1$, ensuring a direct comparison between the two environments. To introduce artificial shifted edges, we randomly selected $0.2 \cdot d$ nodes from the non-root nodes, where $d$ represents the total number of nodes in the DAG. These selected nodes correspond to shifted nodes, denoted as $S$, with $|S| = 0.2d$. For each shifted node $X_j \in S$, we uniformly and randomly deleted 3 edges originating from its parents for environment $\mathcal{E}_2$. The parent nodes whose edges to $X_j$ were deleted are denoted as $D_j$.

The functional relationship between shifted node $X_j$ and its parents $D_j$ in environment $\mathcal{E}_1$ was defined as follows:

$$X_j = \sum_{i \in \mathrm{PA}_j, i \notin D_j} \sin(X_i^2) + \sum_{i \in D_j} c_{ij} \cdot f_{ij}(-2X_i^3 + 3X_i^2 + 4X_i) + N_j,$$

where $c_{ij} \sim \mathrm{Uniform}([-5, -2] \cup [2, 5])$, and $f_{ij}$ is a function from $\{\mathrm{sinc}(\cdot), \cos(\cdot)\}$ chosen uniformly at random. For environment $\mathcal{E}_2$, where the adjacency matrix has undergone deletions, we defined the functional relationship between each node and its parents as follows:

$$X_j = \sum_{i \in \mathrm{PA}_j} \sin(X_i^2) + N_j$$

**Experiment detail.** In each simulation, we generated $\{500, 1000\}$ data points, with the variances of the noises set uniformly at random in $[0.25, 0.5]$. We tested three types of noise distributions, namely, the Normal, Laplace, and Gumbel distributions. We conducted a 100 simulations for each combination of graph type, noise type, and number of nodes. The running time was recorded by executing the experiments on an Intel Xeon Gold 6248R Processor with 8 cores. For our method, we used the hyperparameter $\eta = 0.001$. Different from the hard threshold of $t = 2$ used in previous experiments, we now used the elbow approach to determine the set of shifted nodes. To automatically select the elbow we made use of the Python package `Kneed`[7], with hyperparameters `curve='convex'`, `direction='decreasing'`, `online=online`, `interp_method='interp1d'`.

**Evaluation.** In this experiment, we assessed the performance of our method in two aspects: detecting shifted nodes and recovering the difference DAG. For the evaluation of shifted node detection, we measured F1 score, recall, and precision. In the evaluation of difference DAG recovery, we compared the estimated difference DAG with the ground truth difference DAG using F1 score.

In Figures 17 and 18 we present the performances when using the elbow approach discussed in Remark 6. In Figure 17, we note that iSCAN performs similarly for number of samples 500 and 1000. We also show the top-1, top-2, and top-k precision of iSCAN when choosing the first, first 2, and first k variables of `stats` (see Algorithm 3) after sorting in decresing order, respectively. We remark that the superbly performance of iSCAN in top-1 or top-2 precision suggests that in situations that is difficult to choose a threshold for Algorithm 3, the practioner can consider that the first or first two variables of `stats` are more likely to be shifted nodes. Finally, in Figure 18 we show that iSCAN outperforms DCI in recovering the underlying structural difference.

---

[7]We used the latest version found at: https://kneed.readthedocs.io/en/stable/.

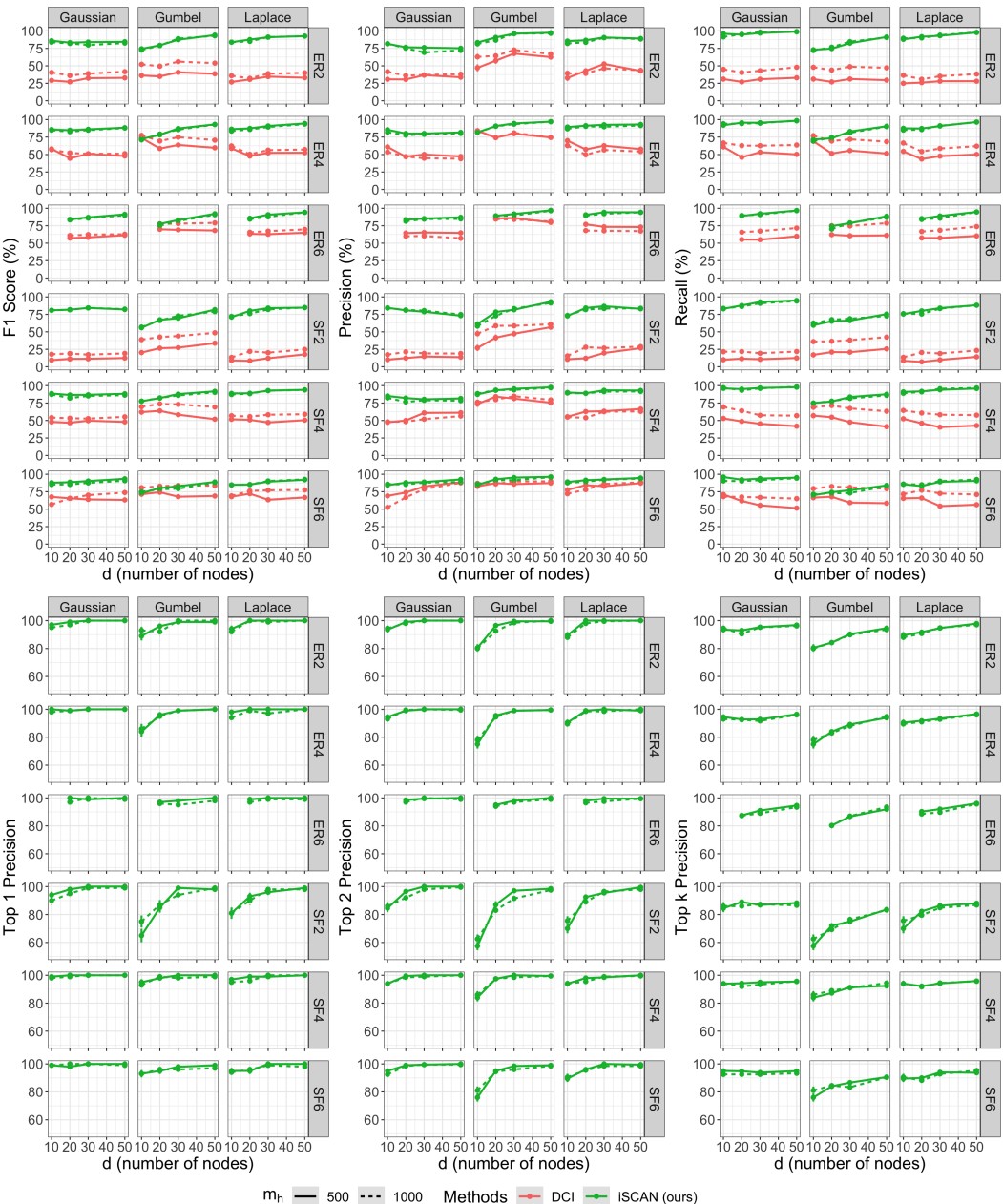

Figure 17: Shifted nodes detection performance in ERk and SFk for $k \in \{2, 4, 6\}$. For each point in each subplot, we conducted 100 simulations as described in Section C.3. The points indicate the average values obtained from these simulations. The error bars depict the standard errors. Our method iSCAN (green) consistently outperformed DCI (red) in terms of F1 score, precision, and recall.

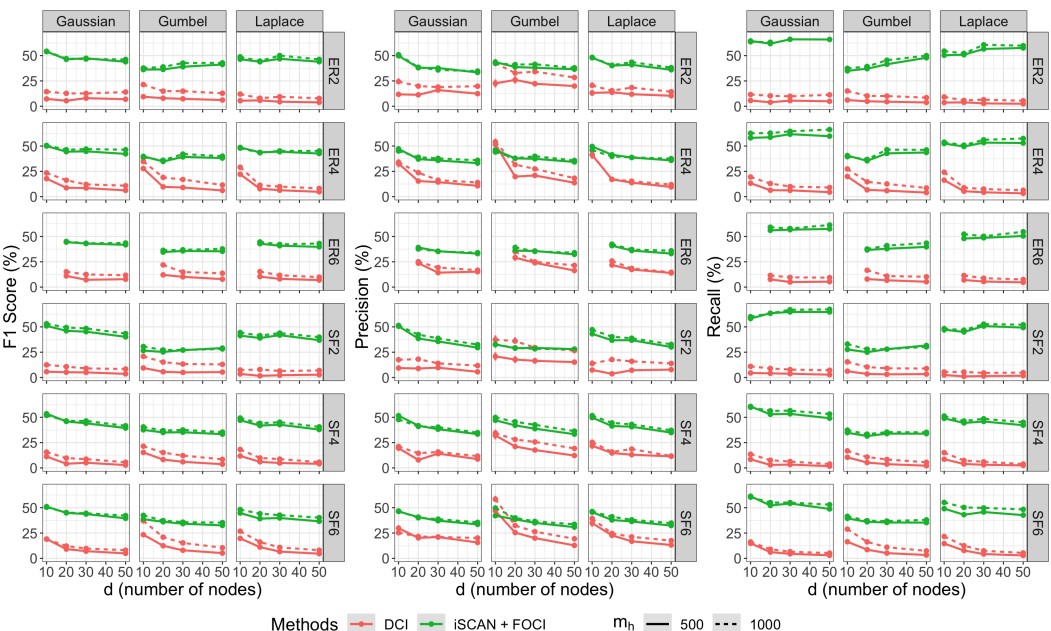

Figure 18: Difference DAG recovery performance in all different graphs. For each point in each subplot, we conducted 100 simulations as described in Section C.3. The points indicate the average values obtained from these simulations. The error bars depict the standard errors. Our method iSCAN with FOCI (green) consistently outperformed DCI (red) in terms of F1 score.

# D   Additional Discussion on Shifted Edges

In Section 4, we focused on estimating structural changes across the environments (Definition 4). However, in some situations it might be of interest to determine whether the *functional relationship* between two variables has changed across the environments. The latter could have multiple interpretations, in this section, we elaborate on a particular type of functional change via partial derivatives.

**Definition 5** (functionally shifted edge). *Given environments $\mathcal{E} = (X, f^h, \mathbb{P}_N^h)$ for $h \in [H]$, an edge $(X_i \to X_j)$ is called a functionally shifted edge if there exists $h, h' \in [H]$ such that:*

$$\frac{\partial}{\partial x_i} f_j^h(\mathrm{PA}_j^h) \neq \frac{\partial}{\partial x_i} f_j^{h'}(\mathrm{PA}_j^{h'}).$$

Without further assumptions about the functional form of $f_j^h$, certain ill-posed situations may arise under Definition 5. Let us consider the following example.

**Example 1.** *Let $\mathcal{E}_A$ and $\mathcal{E}_B$ be two environments, each consisting of three nodes. Let the structural equations for node $X_3$ be: $X_3^A = \exp\left(X_1^A + X_2^A\right) + N_3$, and $X_3^B = \exp\left(2 \cdot X_1^B + X_2^B\right) + N_3$. In this scenario, one could consider that the causal relationship $X_2 \to X_3$ has not changed. However, we note that $\frac{\partial f_3^A}{\partial x_2^A} \neq \frac{\partial f_3^B}{\partial x_2^B}$, thus, testing for changes in the partial derivative would yield a false discovery for the non-shifted edge $X_2 \to X_3$.*

Ill-posed situations such as the above example can be avoided by additional assumptions on the functional mechanisms. We next discuss a sufficient condition where the partial derivative test for functional changes is well-defined.

**Assumption D** (Additive Models). *Let $S$ be the set of shifted nodes across all the $H$ environments. Then, for all $j \in S, h \in [H]$:*

$$f_j^h(\mathrm{PA}_j^h) = a_j^h + \sum_{k \in \mathrm{PA}_j^h} f_{jk}^h(X_k),$$

*where $a_j^h$ is a constant, $f_{jk}^h$ is a nonlinear function, where $f_{jk}^h(\cdot)$ lies in some space of function class $\mathcal{F}$.*

**Remark 7.** *Assumption D amounts to modelling each variable as a generalized linear model [27]. It is widely used in nonparametrics and causal discovery [12, 43, 80]. Moreover, it not only provides a practical framework but also makes the definition of shifted edges (as per Definition 5) well-defined and reasonable.*

**Remark 8.** *Note that Assumption D makes assumptions only on the set of shifted nodes. This is because the set of invariant nodes can be identified regardless of the their type of structural equation, and it is also clear these nodes cannot have any type of shift.*

Now consider a function class $\mathcal{F}$, which incorporates the use of basis functions to model the additive components $f_{jk}^h$. Specifically, we express $f_{jk}^h(x_k) = \Psi_{jk}^h(x_k)\beta_{jk}^h$, where feature mapping $\Psi_{jk}^h$ is a $1 \times r$ matrix whose columns represent the basis functions and $\beta_{jk}^h$ is an $r$-dimensional vector containing the corresponding coefficients. Moreover we assume that the functions $f_{jk}^1, \ldots, f_{jk}^H$ share a same feature mapping $\Psi_{jk}^1(\cdot) =, \ldots, = \Psi_{jk}^H(\cdot)$ but can have different coefficients $\beta_{jk}^h$ across the $H$ environments. The latter has been assumed in prior work, e.g., [44]. The approach of using a basis function approximation is widely adopted in nonparametric analysis, and it has been successfully employed in various domains such as graph-based methods [80], and the popular CAM framework [12]. Then, under Assumption D and Definition 5, we present the following proposition:

**Proposition 2.** *Under Assumption D, an edge $(X_i \to X_j)$ is a functionally shifted edge, as in Definition 5, if and only if the basis coefficients are different. That is,*

$$\frac{\partial f_j^h}{\partial x_i} \neq \frac{\partial f_j^{h'}}{\partial x_i} \iff \beta_{ji}^h \neq \beta_{ji}^{h'}.$$

*Proof.* We have,

$$\frac{\partial f_j^h}{\partial x_i} = \frac{\mathrm{d} f_{ji}^h(x_i)}{\mathrm{d} x_i} = \frac{\mathrm{d}(\Psi_{ji}(x_i)\beta_{ji}^h)}{\mathrm{d} x_i} = \frac{\mathrm{d}\Psi_{ji}(x_i)}{\mathrm{d} x_i}\beta_{ji}^h.$$

Then,

$$\frac{\partial f_j^h}{\partial x_i} - \frac{\partial f_j^{h'}}{\partial x_i} = \frac{\mathrm{d}\Psi_{ji}(x_i)}{\mathrm{d}x_i}(\beta_{ji}^h - \beta_{ji}^{h'}) \neq \mathbf{0} \iff \beta_{ji}^h - \beta_{ji}^{h'} \neq \mathbf{0}$$

The last $\iff$ relation is due to the linear independence of the basis functions $\Psi_{ji}$, then the null space of $\mathrm{d}\Psi_{ji}/\mathrm{d}x_i$ can only be the zero vector $\mathbf{0}$. $\qquad\square$

Note that the output of Algorithm 1 also estimates a topological order $\hat{\pi}$. However, the *exact parents* of a node $X_j$ across the environments are not known, and they are possibly different. To estimate the coefficients without knowledge of the exact parents, we can consider the set $\widehat{\mathrm{Pre}}(X_j)$, which consists of nodes located before $X_j$ in the topological order $\hat{\pi}$. By regressing $X_j$ on $\widehat{\mathrm{Pre}}(X_j)$ for each environment, we can obtain coefficient estimations, which are the same coefficients obtained by regressing $X_j$ on its exact parents, in large samples.

**Theorem 3.** *In large samples, let $\{\tilde{\beta}_{jk}^h\}_{k\in\mathrm{Pre}(X_j)}$ be the coefficients obtained by regressing $X_j$ on the feature mapping of $\mathrm{Pre}(X_j)$, and let $\{\beta_{jk}^h\}_{k\in\mathrm{PA}_j}$ be the coefficients obtained by regressing $X_j$ on the feature mapping of $\mathrm{PA}_j$. Then, $\tilde{\beta}_{jk}^h = \beta_{jk}^h$ if $k \in \mathrm{PA}_j^h$, and $\tilde{\beta}_{jk}^h = 0$ if $k \in \mathrm{Pre}(X_j) \setminus \mathrm{PA}_j^h$.*

*Proof.* Proof can be found in Appendix B.3. $\qquad\square$

Motivated by Theorem 3, and given an estimated $\{\tilde{\beta}_{jk}^h\}_{k\in\widehat{\mathrm{Pre}}(X_j)}$, one could conduct a hypothesis testing as follows:

$$H_0 : \tilde{\beta}_{jk}^1 = \cdots = \tilde{\beta}_{jk}^H \tag{13}$$

If the null hypothesis $H_0$ is rejected, it indicates that there is evidence of a functionally shifted edge between nodes $X_k$ and $X_j$ across the environments. In this paper we leave the hypothesis test unspecified to allow for any procedure that can test eq.(13).

---

**Algorithm 5** Functionally shifted edges detection

---

**Input:** Sample data $\boldsymbol{X}^1, \ldots, \boldsymbol{X}^H$, shifted nodes set $\widehat{S}$, topological order $\hat{\pi}$, significance level $\alpha$.
**Output:** Set of functionally shifted edges $\widehat{E}$
1: **for** $j \in \widehat{S}$ **do**
2:      Estimate $\tilde{\beta}_{jk}^h$ for all $k \in \widehat{\mathrm{Pre}}(X_j)$ and $h \in [H]$
3:      **for** $k \in \widehat{\mathrm{Pre}}(X_j)$ **do**
4:          Conduct hypothesis testing $H_0$ (equation 13) under significant level $\alpha$.
5:          If $H_0$ is rejected, add edge $(X_k \to X_j)$ to $\widehat{E}$.

---

