# OpenReview forum: "iSCAN: Identifying Causal Mechanism Shifts among Nonlinear Additive Noise Models"
_NeurIPS.cc/2023/Conference — NeurIPS 2023 poster_

### Official Review · Reviewer_appw · 2023-06-29

**Soundness:** 3 good
**Presentation:** 2 fair
**Contribution:** 3 good
**Rating:** 6
**Confidence:** 4

**Summary:**

This paper focuses on identifying mechanism shifts in related structural causal models without learning the structures of the individual SCMs. It builds on the recent advances in score-matching based causal discovery and consider nonlinear additive noise models, unlike the prior work that assumed linear Gaussian models. The proposed algorithm first identifies the shifted variables and then estimates the structural changes associated with each shifted variable.

**Strengths:**

- I find the main problem — identifying the changes without learning the graphs in their entirety — important.
- Score-based (gradient of log likelihood) are gaining momentum in causal inference and this can be a timely paper to expand the existing recent work. Also, I think soft interventions are also relatively understudied but are important for modeling different environments with sparse changes.
- Paper is written clearly, the flow is natural and easy to follow.

**Weaknesses:**

- *Invariant noise assumption (Assumption B) and shifted node definition*: I see that invariant noise assumption is crucial for your results, especially it is used in Lemma 2 (and subsequently for the proof of Theorem 1). However, I find this a somewhat unrealistic assumption. For instance, your current definition of shifted nodes excludes root nodes — they can be intervened only upon a change in the noise distribution. I believe Assumption B needs a convincing discussion. I add more comments on questions below.
- The main contribution (Theorem 1) is rather a straightforward extension of the existing result (Proposition 1). Sure, there may be technical challenges, but I don’t see a fundamental difficulty. If I am missing anything, please let me know.
- Although the paper provides extensive experiments in numerous settings, it misses comparisons with some important related work.

========= AFTER AUTHORS' REBUTTAL ==========
A major concern, the invariant noise assumption, is addressed by the reviewers in their rebuttal. Hence, I updated my score accordingly.

**Questions:**

- After finding the shifted nodes, finding the parents should not be a big deal besides the computational burden. One angle can be your efficiency of Algorithm 2. Can you elaborate on its efficiency and which parts are your contribution?
- Invariant noise (Assumption B): How do you deal with the root nodes? Do you simply assume that root nodes are never intervened? I see that this is how simulations are conducted. It should be clearly stated in the paper as well, as part of the assumptions perhaps.
- If I recall correctly, DCI algorithm of Wang et al. [77] does not require invariant noise assumption. In contrast, the algorithm of Ghoshal et al. (2019) [23] relies on invariant noise and offers some advantages over DCI. I believe the algorithm of Ghoshal et al. can be an important comparison too.
- Speaking of related work and comparisons, I believe the present work misses an important reference, “Scalable intervention target estimation in linear models” Varici el al. (2021). Although it’s focused on just identifying the intervened (shifted) nodes, their algorithm also finds some edges. For the comparison of Algorithm 1, along with Ghoshal et al. (2019), Varici et al. (2021) should also be a valid comparison in addition to the already used DCI. The formulation in Varici et al. (2021) differs by assuming that the noise variables necessarily change, but I believe a change in the linear weights is enough to relax the “varying noise” assumption. (you can check their follow-up Varici et al. (2022) in which they discuss the same problem for causally insufficient models with multiple environments)
- Finally, all these three papers (Wang et al., Ghoshal et al., Varici et al.) are based on linear models whereas the present paper requires non-linear models. More complex algorithms that can work with non-parametric causal models can be used. For instance, UT-IGSP algorithm of Squires et al. (2020), JCI algorithm of Mooij et al. (2020), and $\psi$-FCI algorithm of Jaber et al. (2020) all can return intervention targets along with the causal graph. The latter two can be computationally prohibitive, but I think UT-IGSP can work on fairly large graphs as well for a fair comparison of identifying shifted nodes.
- Including code in supplementary could have been helpful.

 Varici, B., Shanmugam, K., Sattigeri, P., & Tajer, A. (2021). Scalable intervention target estimation in linear models. Advances in Neural Information Processing Systems, 34, 1494-1505.

Varici, B., Shanmugam, K., Sattigeri, P., & Tajer, A. (2022, August). Intervention target estimation in the presence of latent variables. In Uncertainty in Artificial Intelligence (pp. 2013-2023). PMLR.

Squires, C., Wang, Y., & Uhler, C. (2020, August). Permutation-based causal structure learning with unknown intervention targets. In Conference on Uncertainty in Artificial Intelligence (pp. 1039-1048). PMLR.

Mooij, J. M., Magliacane, S., & Claassen, T. (2020). Joint causal inference from multiple contexts. The Journal of Machine Learning Research, 21(1), 3919-4026.

Jaber, A., Kocaoglu, M., Shanmugam, K., & Bareinboim, E. (2020). Causal discovery from soft interventions with unknown targets: Characterization and learning. Advances in neural information processing systems, 33, 9551-9561.

**Minor notes**:

- Montagna et al. “Scalable Causal Discovery with Score Matching”: This is cited in the appendix. I believe it deserves more discussion in the main paper as it has some improvements over Rolland et al. (2020) especially for parent selection and efficiency of their algorithm. I see that the recent version became public at arXiv just shortly before the submission of the present paper, though an earlier version was available (NeurIPS 2022 Workshop on Score-Based Methods).
- *ICM assumption*: Do you really need ICM assumption? It seems to me that you only need modularity principle of Pearl, that is each causal mechanism (conditional prob. dist.) is independent of the others. I see that you use it to say that “there are only a few shifted nodes” but if I am not mistaken, your analysis does not depend on such sparsity assumption.
-  [62] (Saeed et al. 2020): To my knowledge, their result for identifying shifted variables comes under poset compatibility assumption that can be quite restrictive for the structural changes.


**Limitations:**

The main limitations of the present work would be their assumptions. They are not discussed in a separate paragraph or pointed out somewhere in the paper (if I am not missing). But they are clearly written and can be understood at where they appear in the main text.

---

> ### Author Rebuttal · Authors · 2023-08-10
>
> Thanks for your valuable feedback! We next respond to your questions, and hope our answers will make you feel more positive about our work. Please do not hesitate to ask any follow-up questions.
>
> ### Response to comments from Weaknesses section
>
> > "Invariant noise assumption (Assumption B) and shifted node definition..."
>
> We have great news regarding Assumption B, we can relax it and allow for changes in the noise distribution! Our method is now capable of identifying mechanism shifts of a node, stemming from either changes in the functional relationship w.r.t. its parents or shifts in the noise distribution. Please see our rebuttal to all reviewers for more details.
>
> > "The main contribution (Theorem 1) is rather a straightforward extension of the existing result (Proposition 1). Sure, there may be technical challenges, but I don’t see a fundamental difficulty. If I am missing anything, please let me know."
>
> We respectfully disagree with your comment, but we earnestly hope to clarify our perspective and appeal to your understanding.
> Our work's novelty can be seen from two angles: the question we address and the method we propose.
>
> * The Question: As you kindly acknowledge in the strengths, the problem we tackle is both interesting and important. To the best of our knowledge, this paper is the first to consider this specific question, a natural extension motivated by previous work such as DCI. We genuinely believe that coming up with a more efficient algorithm that is different from the naive approach (i.e., estimating individual DAGs, fitting functionals, and testing for differences) is already a novel contribution.
>
> * The Method: First, our proposed method is innovative in that it bypasses any parametric form assumption on the functional mechanisms, and does not rely on any structural assumption in the individual DAGs, as pointed out in our contributions (L63-L70). The effectiveness of our approach is demonstrated through empirical experiments, and it is grounded on solid theory. Second, the proof of Theorem 1 is very different and more convoluted than the proof of Proposition 1 from Rolland et al. Finally, Theorem 1 is easy to digest and our algorithm is simple to implement, making our paper more accessible for others to understand and apply. We sincerely hope you can see that the simplicity of our approach should not be mistaken for a lack of innovation.
>
>
> ### Response to comments from Questions section
>
> > "After finding the shifted nodes, finding the parents should not be a big deal besides the computational burden. One angle can be your efficiency of Algorithm 2. Can you elaborate on its efficiency and which parts are your contribution?"
>
> We argue that finding the parents of a node (a.k.a. Local causal discovery) is also a very challenging task, not just computationally but also methodologically; see for instance Gao & Aragam (2021) “Efficient Bayesian network structure learning via local Markov boundary search”. Our approach addresses this task by using an estimated topological ordering returned by Alg 1, and adapting recent methods like FOCI, which efficiently measures the level of dependence between random variables. We've shown FOCI's efficiency empirically and its consistency for local recovery in Theorem 2.
>
>
> > "If I recall correctly, DCI algorithm of Wang et al. [77] does not require invariant noise assumption..."
>
> To be precise *both DCI and Ghoshal et al.’s method assume invariant noises in order to identify all edge orientations.* Without such assumption both methods can only output a partially oriented difference DAG. See Theorem 4.4 in Wang et al.’s DCI paper, and Theorem 1 in Ghoshal et al.’s paper.
> Regarding a comparison against Ghoshal et al.'s method. We should point out that, as illustrated in Fig 3 of their paper, the difference in performance against DCI is not significant. In addition, their code is not available. Given that both DCI and Ghoshal et al.'s study linear models, we believe that DCI is sufficient as a baseline.
>
>
> > "Speaking of related work and comparisons, I believe the present work misses an important reference..."
>
> Thank you for suggesting these references! We will make sure to include them in the revised version. Additionally, we have incorporated some additional baselines in our experiments, as detailed in our rebuttal to all reviewers.
> It is also worth noting that *our problem setting is not the same as identifying intervened nodes.* Please see our response to all reviewers.
>
>
> > "Finally, all these three papers (Wang et al., Ghoshal et al., Varici et al.) are based on..."
>
> Thank you again for sharing additional related work. We have included UT-IGSP as a baseline as well, see our response to all reviewers.
>
>
> > Including code in supplementary could have been helpful.
>
> We will make sure to release code for the camera ready.
>
>
> > "Montagna et al. “Scalable Causal Discovery with Score Matching”: This is cited in the appendix..."
>
> We will certainly discuss this work in our revised version. However, from our understanding, Montagna et al.’s work does not improve on the efficiency, nor in parent selection. It cannot be more efficient as they also perform a score estimation step as in Rolland et al.’s, and the edge pruning step (or parent selection) is based on CAM’s pruning procedure. On the other hand, Montagna et al.’s work does support arbitrary noise distributions, which makes for intriguing future work to investigate how to use some of their ideas for our problem setting.
>
>
> > "ICM assumption..."
>
> Thanks for pointing this out. Indeed, we only need the modularity principle, we will correct this in the revision.

---

> > ### Comment · Reviewer_appw · 2023-08-11
> >
> > I thank the authors for the rebuttal, it answered my major questions. Specifically, relaxing Assumption B addresses my biggest concern. Adding more experimental comparisons is certainly more convincing too. I will update my review and score soon to reflect these.
> >
> > It can be a minor nitpick, but I want to clarify one final thing. In the third point of the global response, *".. not all intervened nodes are shifted nodes. .. For example, it is possible that a variable has the same intervention across the environments, and thus, it would not be considered a shifted node... "*
> >
> > I guess we have a slightly different approach to defining what is an intervention. If one has two environments in which a node shares the same mechanism (i.e., no shift), of course, the goal of "identifying interventions"  would not be to return this node in the sense that an interventional mechanism should be defined as opposed to an observational one. In that case, the node's mechanism can be considered observational, even though physically it's interventional, simply because there is only one mechanism and nothing to distinguish about. With this perspective, the problem of intervention target estimation (under the proper problem definition) is equivalent to identifying mechanism shifts. Please correct me if I am missing something regarding your work and assumptions.

---

> > > ### Comment · Area_Chair_mwzv · 2023-08-18
> > >
> > > Dear Authors,
> > >
> > > since the discussion period is soon coming to an end, can you please respond to the remaining open comments from reviewer appw (and reviewer PS2y)?
> > >
> > > All the best,
> > >
> > > Your AC.

---

> > > ### Author Response · Authors · 2023-08-18
> > >
> > > Dear reviewer, we are glad to see our responses were helpful and that we were able to address your primary concern!
> > >
> > > Also, thank you for your insightful comment! We agree that *in the case of two environments*, both problems could be considered equivalent *in some cases*. However, we would like to note the following:
> > >
> > > * **On the semantics.** When one has access to more than 2 datasets, the language could be confusing if one does not properly distinguish an intervention from a mechanism shift. For example, suppose we have 3 environments $\\{\mathcal{E}_0, \mathcal{E}_1, \mathcal{E}_2 \\}$, where environment $\mathcal{E}_0$ corresponds to the observational distribution. Also, suppose node $i$ has the following structural equations:
> > >   * In $\mathcal{E}_0$,  we have $X^{(0)}_i = \sin(X_j) \cdot \cos(X_k) + N^{(0)}_i$, with parent set $\mathrm{PA}_i^{(0)} = \\{j,k\\}$.
> > >   * In $\mathcal{E}_1$, we have $X^{(1)}_i = \sin(X_j) +  N^{(1)}_i$, with parent set $\mathrm{PA}_i^{(1)} = \\{j\\}$.
> > >   * In $\mathcal{E}_2$, we have $X^{(2)}_i = \sin(X_j) + N^{(2)}_i$, where $N^{(1)}_i$ and $N^{(2)}_i$ have the same distribution, and with parent set  $\mathrm{PA}_i^{(2)} = \\{j\\}$.
> > >
> > >   In this example, the mechanisms of $X_i$ are the same in environments $\mathcal{E}_1$ and $\mathcal{E}_2$ even though $X_i$ is an intervention target in both environments w.r.t. the observational environment $\mathcal{E}_0$.
> > >
> > > * **On the assumptions.** We also point out that prior works that consider unknown intervention targets typically aim to estimate the $\mathcal{I}$-MEC and, thus, rely on different (and in many cases stronger) assumptions. To give a concrete example, in Squires et al. (UT-IGSP), the authors argue about the necessity of "direct $\mathcal{I}$-faithfulness" (Assumption 1 in their paper) in order to identify the $\mathcal{I}$-MEC, which we do not need to assume.
> > >
> > >   Also, it is important to note that prior works that aim to estimate the $\mathcal{I}$-MEC rely on knowing which of the provided datasets corresponds to the observational distribution in order to have well-defined (soft or hard) interventions. Relating to the example in the previous bullet, consider that we only have access to 2 datasets generated from $\mathcal{E}_0$ and $\mathcal{E}_1$, and **suppose we do not know which corresponds to the observational distribution**. Then, if we choose $\mathcal{E}_1$ as the "observational distribution", the other environment $\mathcal{E}_0$ would correspond to the "interventional" distribution where $X_i$ has been intervened by **adding** a new causal parent $X_k$, thus, violating the usually assumed definition of soft/hard intervention. In contrast, our approach does not require knowledge of which dataset corresponds to the observational distribution.
> > >
> > > Hope this clarifies your comment! We believe this discussion is important so we will add a short paragraph in the revision to remark these points. Please let us know if you have any other comments!

---

### Official Review · Reviewer_PS2y · 2023-07-03

**Soundness:** 2 fair
**Presentation:** 2 fair
**Contribution:** 2 fair
**Rating:** 5
**Confidence:** 3

**Summary:**

The authors propose a two-step algorithm for identifying causal mechanism shifts. The algorithm is motivated by a characterization of the nodes with mechanism shifts and the procedure is based on several existing tools. The proposed algorithm outperforms the baseline.

**Strengths:**

1. Theorem 1 provides a new perspective for the characterization of nodes with shifted mechanisms.

2. The combination of existing tools in Algorithm 2 is interesting.

3. The computation complexity of the algorithm is reported.

**Weaknesses:**

1. Assumption B (invariant noise distribution) makes the setting quite restrictive. There is no discussion or experiments about how sensitive the proposed method is w.r.t. shifted noise distributions, or whether the proposed idea can be generalized to handle shifted noise distributions.

2. I suspect that the statistic is not well-defined in some cases (see Questions below).

3. The threshold $t$ is a key hyper-parameter for the method, but the suggested elbow strategy is not a principled approach.



**Questions:**

Regarding the statistics defined in (6) from Appendix, if the leaf node L is not shifted, the statistics may not be small since both the numerator and denominator are approximately zeros. Did I miss something here?






**Limitations:**

No potential negative societal impact.

---

> ### Author Rebuttal · Authors · 2023-08-10
>
> Thanks for your comments! We next respond to your concerns, and hope our answers will make you feel more positive about our work. Please do not hesitate to ask any follow-up questions.
>
> > Assumption B (invariant noise distribution) makes the setting quite restrictive. There is no discussion or experiments about how sensitive the proposed method is w.r.t. shifted noise distributions, or whether the proposed idea can be generalized to handle shifted noise distributions.
>
> We have excellent news here, we can drop Assumption B and still have our method to identify distribution shifts without major changes to our proof of Theorem 1! Please see our response to all reviewers. Given that your main concern is related to this assumption, we hope you will feel more positive about our contributions.
> > The threshold t is a key hyper-parameter for the method, but the suggested elbow strategy is not a principled approach.
>
> We agree but note that we never claimed that the elbow method is a principled approach, in fact, it is well known to be a classical heuristic for estimating the number of clusters in a dataset. In this case, as the experiments in Appendix D.3 suggest, it can lead to good performances.  Finally, we also believe that a principled approach would be nice to have, which makes this an exciting future direction!
>
>
> > Regarding the statistics defined in (6) from Appendix, if the leaf node L is not shifted, the statistics may not be small since both the numerator and denominator are approximately zeros. Did I miss something here?
>
> Great question! If the score estimator is well behaved (meaning that as the number of samples increases, the estimated value gets closer to its population value), then, for a non-shifted leaf node $L$, the numerator would converge faster towards zero. This is due to the numerator using all the pooled data for the estimation of the score; on the other hand, the denominator uses only data from a single environment. Thus, when the leaf node $L$ is not shifted, we can expect the statistic $\mathtt{stats_L}$ to be lower than 1.

---

> > ### Comment · Reviewer_PS2y · 2023-08-16
> > **Reply to rebuttle**
> >
> > It is good to see that Assumption B can be relaxed. Based on this, I would raise my score.
> >
> > If the elbow strategy is indeed a classic heuristic, please provide references to support it.
> >
> > For the statistics, the intuition sounds reasonable.  But since it is an important component of the method, some rigorous results will make the method more convincing.

---

> > > ### Author Response · Authors · 2023-08-21
> > >
> > > Dear reviewer, thank you for your comment. We are glad to see that our response made you feel more positive about our work.
> > >
> > > * Regarding the elbow method, the idea can be traced back to [1]. We are including a few other references that use or explain this heuristic [2,3,4]. We should point out that this elbow approach was merely tested as an alternative to the threshold approach, and its inclusion in our work was based on the reasonable performance we observed using this method.
> > >
> > > * As per your comment regarding "some rigorous results", we would like to remind you that our theory on the identifiability of mechanism shifts is rigorous and all the proofs are given in the appendix. Since our theory is given at the population level, we rely on algorithmic approaches or having a hyperparameter (the threshold $t$) in order to implement our method in finite samples. This is not unique to our work, in fact, even Rolland et al.'s SCORE method uses a heuristic to choose a leaf from a set of nodes---namely, pick the node as the diagonal entry with the lowest sample variance of the score's Jacobian. In the context of causal discovery, even prominent algorithms such as the PC or GES algorithms are proved at the population level and rely on different hyperparameters to work in finite samples, e.g., sparsity coefficient, and alpha level.
> > >
> > >   Finally, *one way to obtain a principled threshold for our algorithm would require an estimator of the score function with finite-sample guarantees*. Such a result can make a standalone research paper, as the utility of score matching spans various domains, encompassing generative and discriminative models [5,6,7,8,9], and more recently, applications in causal discovery [10] and causal representation learning [11].
> > >
> > > *We hope the notes above will help clarify your comments.*
> > >
> > >
> > >
> > > [1]: Thorndike, R. L. (1953). "Who belongs in the family?." Psychometrika.
> > >
> > > [2]: V. Satopaa, J. Albrecht, D. Irwin and B. Raghavan. (2011). "Finding a "Kneedle" in a Haystack: Detecting Knee Points in System Behavior." 31st International Conference on Distributed Computing Systems Workshops.
> > >
> > > [3]: Goutte, C., Toft, P., Rostrup, E., Nielsen, F. Å., & Hansen, L. K. (1999). "On clustering fMRI time series". NeuroImage.
> > >
> > > [4]: Dangeti, P. (2017). Statistics for machine learning. Packt Publishing Ltd.
> > >
> > > [5]: Song, Y., & Ermon, S. (2019). Generative modeling by estimating gradients of the data distribution. Advances in neural information processing systems, 32.
> > >
> > > [6]: Zimmermann, R. S., Schott, L., Song, Y., Dunn, B. A., & Klindt, D. A. (2021). Score-based generative classifiers. arXiv preprint arXiv:2110.00473.
> > >
> > > [7]: Song, Y., Sohl-Dickstein, J., Kingma, D. P., Kumar, A., Ermon, S., & Poole, B. (2020). Score-based generative modeling through stochastic differential equations. arXiv preprint arXiv:2011.13456.
> > >
> > > [8]: Song, Y., Garg, S., Shi, J., & Ermon, S. (2020, August). Sliced score matching: A scalable approach to density and score estimation. In Uncertainty in Artificial Intelligence (pp. 574-584). PMLR.
> > >
> > > [9]: Song, Y., & Ermon, S. (2020). Improved techniques for training score-based generative models. Advances in neural information processing systems, 33, 12438-12448.
> > >
> > > [10]: Rolland, P., Cevher, V., Kleindessner, M., Russell, C., Janzing, D., Schölkopf, B., & Locatello, F. (2022, June). Score matching enables causal discovery of nonlinear additive noise models. In International Conference on Machine Learning (pp. 18741-18753). PMLR.
> > >
> > > [11]: Varici, B., Acarturk, E., Shanmugam, K., Kumar, A., & Tajer, A. (2023). Score-based causal representation learning with interventions. arXiv preprint arXiv:2301.08230.

---

> > > > ### Comment · Reviewer_PS2y · 2023-08-21
> > > > **Reply**
> > > >
> > > > I am fine with the elbow strategy if it can work well in practice.
> > > >
> > > > Apparently, I am saying that there are no rigorous results regarding the statistics. Talking about population guarantees, the statistics is not well-defined, since it becomes zero divided by zero when the leaf node L is not shifted. Note that "there is a leaf node L that is not shifted" is very common scenario. The algorithms you mentioned are well-defined in the population case. I am not asking you to provide finite sample analysis. I think it is better to propose a new statistics that is correct.

---

> > > > > ### Author Response · Authors · 2023-08-21
> > > > >
> > > > > Thanks for the clarification! In that case, it simply suffices to add a small epsilon value to the denominator to avoid the zero issue, i.e., $\mathsf{stats}_L = \frac{\mathrm{Var}_L}{\min_h \mathrm{Var}_L^h + \varepsilon}$, where, e.g., $\varepsilon = 10^{-12}$. Then in population, the statistic will be $0$ when a leaf node is non-shifted. We will add this small $\varepsilon$ to the definition for further clarity.

---

### Official Review · Reviewer_YGJy · 2023-07-04

**Soundness:** 4 excellent
**Presentation:** 3 good
**Contribution:** 2 fair
**Rating:** 5
**Confidence:** 4

**Summary:**

This work addresses the problem of estimating differences in the causal models in two (or more) environments, over a single set of variables with the same causal ordering. The authors are interested in discovering those variables that undergo a functional or structural mechanism shift. As their main insight, they show that under additive noise models, the Jacobian of the score function of the data distribution provides information about such shifts. With this, they follow up on a line of research that uses this Jacobian to identify leaf nodes in a causal DAG (Rolland et al. 22, Montagna et al. 23).

The authors propose the MSG algorithm, which given data from two environments, first discovers nodes which undergo a mechanism shifts, and in a second stage discovers the causal parents of shifted nodes as well as the edge differences between the causal DAGs in each environment.  Empirically, they evaluate MSG on  discovering nodes with mechanism shifts, and show that MSG outperforms the DCI approach (Wang et al. 18)  while scaling linearly in the number of variables.

**Strengths:**

- *Presentation*: The authors motivate the problem well and place the approach appropriately into the context of existing work. The paper has a clear structure and is easy to follow.
- *Soundness*: The authors state the theoretical results clearly and provide detailed proofs for each claim.
- *Scalability*: As the authors point out, their approach can identify mechanism shifts while avoiding a computationally costly estimation of the complete causal DAG. In addition, the proposed algorithm scales linearly in the number of DAG nodes, contrary to most existing work. Lastly, they propose to estimate the score function using the mixture distribution over multiple environments, suggesting an improved sample efficiency compared to approaches that use per-environment distributions.
- *Assumptions*: MSG does not make any structural assumptions on the DAGs, nor restrictive  functional modelling assumptions.
- *Contribution*: Existing work on discovering difference DAGs assumes linear  functional models (Wang et al. 18, Ghoshal et al. 21). Meanwhile, nonlinear approaches that can indirectly solve this problem by discovering a causal DAG over multiple contexts and identifying mechanism changes for each node (e.g. Mooij et al. 16, Huang et al. 19) do not scale well in practice. Hence, this work is a valuable addition to the literature.


**Weaknesses:**

- *Novelty*: This work follows a recent line of research which also uses the Jacobian of the score of the data distribution to identify leaves in a causal DAG  (SCORE, Rolland et al. 22, Montagna et al. 23). The proposed approach applies similar techniques and algorithms, making the approach incremental in nature.
- *Comparison to SCORE*: To demonstrate the practical relevance of their insights and the novelty compared to the SCORE framework, it would be helpful to see a comparison to SCORE in the experiments. For example, is there a noticeable benefit of MSD compared to applying SCORE to each environment and comparing the resulting DAGs for structural differences?
- *Comparison to DCI*: Given that DCI makes a linearity assumption, the superior performance of MSD in the experiments might only be due to DCI's model misspecification in the nonlinear setting. It might be interesting to include the linear case, where MSD's functional model is misspecified, or to consider additional competitors.
- *Competitors:* There are many approaches that solve this problem indirectly, by (a) discovering a joint DAG over multiple environments and functional shifts for each node, or (b) discovering a DAG per environment, which we can compare for  structural differences. For discovering shifted nodes, MSG has a computational advantage of not needing a direct estimation of DAGs, but it would be interesting to see whether it also has an advantage in terms of accuracy.  For discovering structural differences, MSG has an F1 score of around 0.5 (Fig. 11), thus it would be informative to see whether direct DAG estimation improves upon this.



**Questions:**

- What is the best way to extend MSG to more than two environments? For example, is it still possible to estimate the score from the mixture distribution over multiple contexts, and if so, how does this depend on the number of mechanism shifts over all contexts?
- Definition 3 seems to admit the case where the causal parent set is empty in a given environment, that is, where a hard intervention occurs. Given that this case is not considered in the paper (footnote 3), what problems does it cause for MSD? Regardless, it would be interesting to consider hard intervention in the experiments.
- The authors mention the approach by Ghoshal et al. 21, which is more recent than and slightly outperforms DCI. Is there a reason that the experiments do not include it?
- Why is the evaluation result for 10 nodes sometimes missing in Appendix D (Fig. 10, 11)? I would also be interested in the case of 10 nodes in the main experiment in  Fig. 3, since it seems that MSD has lower precision on smaller graphs (e.g. 20 nodes).
- Fig. 3 seems to suggest that shifted node discovery improves with increasing number of nodes d, whereas in Fig. 1 it degrades, why is this?
- In the experiments on difference DAG discovery in Fig. 11,  MSG has F1 scores of at most 0.5. What most likely causes this? For example, is it related to the estimation of the causal parent sets with Alg. 2? While Theorem 2 establishes the consistency of discovering the correct parent sets, is this also true in practice?
- Can the hypothesis tests in Appendix E be extended to also accomodate changes in noise distributions?
- *smaller comments*: Citation style in Section 2.2 (numbers as subjects), Precision and Recall missing in Fig. 11 (while included in Fig. 13 which has the same setting).



**Limitations:**

- *Types of Mechanism Shifts*: The setting is motivated in the context of both functional and structural differences in SCMs (Def. 3, 4). While Alg. 1 for discovering shifted nodes can deal with both of these shifts, the evaluation focuses only on a specific type of *functional* shifts. Alg. 3 for discovering the causal parents, on the other hand, is only evaluated for *structural* shifts (Appendix D). The authors  discuss the theoretical aspects of using Alg. 3 under functional shifts in Appendix E, but it would be interesting to also see an evaluation.
- *Experiments*: As elaborated under weaknesses, the evaluation is limited because the authors only consider the linear competitor DCI, as well as do not evaluate parts of the proposed algorithm (Alg. 2, Alg. 5) or only in specific settings (Alg. 1, Alg. 3).

---

> ### Author Rebuttal · Authors · 2023-08-10
>
> Thank you for your valuable input. We have new experiments on additional settings, and other baselines, as per your suggestions. Please see response to all reviewers.
>
> ### Response to comments from Weaknesses section
>
> > "Novelty: This work follows ..."
>
> We firmly disagree with the statement that “our approach is incremental in nature”. Here we provide some points to take into account.
>
> 1. *The problem itself is very different to that of Rolland et al.*, and coming up with a more efficient algorithm that is different from the naive approach (i.e., estimating individual DAGs, fitting functionals, and testing for differences) is far from trivial. The effectiveness of our approach is demonstrated through empirical experiments, and it is grounded on solid theory.
>
> 2. The proof of Theorem 1 is very different and more convoluted than the proof of Proposition 1 from Rolland et al.
>
> 3. There is a consensus from the reviewers that the identifiability result of Thm 1 is easy to digest. The latter does not necessarily imply a lack of novelty or creativity; on the contrary, it often means that the method is more accessible for others to understand and apply.
>
> We sincerely hope that you will reconsider our method's novelty. We believe a simple yet effective approach should be celebrated, not criticized.
>
>
> > On additional comparisons
>
> We tested additional settings such as functions drawn from Gaussian processes, and other baselines based on indirect methods. Please see response to all reviewers.
>
> ### Response to comments from Questions section
>
> > "What is the best way to extend MSD to more than two environments?..."
>
> * The problem setting, as stated in L141, considers data from $H$ environments. All the theoretical statements, including proofs, are also given in full generality, i.e., considering $H$ environments. For the experiments, we mainly consider $H=2$ because DCI only works for two datasets.
>
> * Regarding estimation, in L5 of Algorithm 1 (or L9 of Alg 4), we estimate the diagonal of the score’s Jacobian of the mixture distribution, which consists of data gathered from all the environments. This estimation process does not depend on the number of mechanism shifts.
>
>
> > "Def 3 seems to admit ..."
>
> Good catch! We will assume that you meant Def 2, as Def 3 is related to shifted nodes. Correct, our results also support the case where the parents of a variable are completely removed. Formally, our identifiability result also holds for stochastic hard interventions, i.e., the parents are removed but we keep the noise to have a well-defined ANM for each environment.
>
>
> > "...authors mention the approach by Ghoshal et al..."
>
> The reason was primarily because DCI's code was accessible, and Ghoshal et al.'s was not. However, we should point out that, as illustrated in Fig 3 of Ghoshal et al., the difference in performance is not significant. Given that both methods study linear models, we believe that DCI was sufficient as a baseline.
>
> > "Why is the evaluation result for..."
>
> Good catch! The absence is due to not being able to generate graphs with the given specification, in this case, ER6/SF6 graphs. For clarity, at $d=10$, an ER6/SF6 graph corresponds to a graph with 60 edges in expectation, which is impossible to generate since there are at most 45 edges in a DAG of 10 nodes. The reason there exist a point for $d=10$ in SF6 graphs (e.g. in Figs 10, 11) is because the package we used to generate SF graphs automatically sets to the maximum number of possible edges if larger numbers are given; whereas the package to generate ER graphs, does not. We will remove all points for ER6/SF6 graphs at $d=10$.
>
> > "Fig. 3 seems to suggest..."
>
> There should be a typo in your question since Fig 1 does not show any empirical results, thus, we will assume that you refer to Fig 6, not Fig 1.  A high-level explanation can be given by looking at the data generation process in L262-L268. In this setting, the structures in the two environments are exactly the same, and only the functional mechanisms of the shifted nodes change. Then, for a given number of nodes $d$, the mechanisms of shifted nodes in ER6/SF6 graphs will observe higher “fluctuations” since the expected number of parents is higher than that of ER2/SF2 graphs, which our method successfully captures.
>
>
> > "In the experiments on difference DAG..."
>
> Good question. To estimate a difference DAG (Alg 3) we use: Alg 1 to detect the shifted nodes, and Alg 2 for estimating local parents. While both Alg 1 and Alg 2 affect the performance of estimating the difference DAG, we attribute your observation primarily to Alg 2. While Thm 2 provides the consistency result of Alg 2, it may not always translate to strong performance in finite samples. Finally, it is clear that incorrect estimates of shifted nodes from Alg 1 will also affect the final difference DAG accuracy, but we believe Alg 1 is not the main source of errors as it performs quite well as seen in the experiments.
>
>
> > "Can the hypothesis tests..."
>
> See response to all reviewers. We can drop Assumption B, i.e., our method will also identify changes in noise distributions.
>
> ### Response to comments from Limitations section
>
> > "Types of Mechanism Shifts..."
>
> Our paper's primary goal is to identify mechanism shifts without specific functional or structural assumptions, as stated in our contributions (L63-L70). While our method can identify structural and functional shifts in nodes, estimating shifted edges may require an additional set of assumptions. In App E we explored a potential semi-parametric assumption to identify functional shifted edges, but this is not central to our work, and was provided mainly to motivate future directions.
>
> > "Experiments..."
>
> * See new experiments in the response to all reviewers.
> * We executed Alg 3 (and thus Alg 2) in the process of recovering the difference DAG, as detailed in Section D.2. Given that App E is not central to the goal of this paper, we decided not to evaluate Alg 5.

---

> > ### Comment · Reviewer_YGJy · 2023-08-18
> >
> > I appreciate the rebuttal by the authors. Addressing the key comments will require a very major revision of the manuscript, but expecting the authors to do so diligently, I'm willing to update my score to a 5.

---

> > > ### Author Response · Authors · 2023-08-21
> > >
> > > Dear reviewer, thank you for your comment. We are happy to see our responses were helpful and made you feel more positive about our work. However, we argue that our paper does not require "a very major revision". Concretely:
> > >
> > > * _For the main text_, our changes include: (1) Removing Assumption B; (2) Updating the definition of shifted nodes (**by 1 line**) to include changes in the noise distribution (due to removing Assumption B); (3) Theorem 1 will only change from "under Assumptions B, C, and D" $\to$ "under Assumptions C and D", i.e., **only 1 letter is being removed from the statement**; (4) Assumption A, is being updated from ICM to Modularity; and (5) Figure 6 is being replaced to include the other baselines. Overall, we do not consider these changes to be a major revision.
> > >
> > > * _For the appendix_, our changes include: (1) Short updates in the proof of Theorem 1 and Lemma 2 (as stated in our global rebuttal); (2) All figures in the appendix are updated to include new baselines; and (3) Two additional experimental settings will be added---one where Gaussian processes are used to sample functionals, and the other where functions are sampled from a predefined set.
> > >
> > > We hope these points help clarify your concern about the changes for the revision.

---

### Official Review · Reviewer_m244 · 2023-07-04

**Soundness:** 3 good
**Presentation:** 4 excellent
**Contribution:** 3 good
**Rating:** 7
**Confidence:** 4

**Summary:**

The paper proposes an efficient method to identify causal mechanism shifts across various interventional environments. The method's objective is to identify changes not only in the mechanism itself but also in the causal structure across the environments, which has been evaluated in artificial and real-world data sets.

**Strengths:**

* Excellent introduction and illustrative examples to familiarize readers without strong backgrounds in this field with the problem setting.
* Notation is concisely introduced, and the assumptions are detailed thoroughly.
* The work is a theoretically valuable extension of existing work.
* The method is easily applicable, and the details are sufficiently outlined for implementation.
* The complexities are discussed in the paper.

Please see the questions section for additional points.

**Weaknesses:**

* The paper primarily builds on existing work, but it extends them in a sufficiently novel manner. One of my concern is the claim of not requiring knowledge about the causal structure, which is solely due to the utilization of Rolland et al.‘s existing work that aims at identifying the structure. Thus, one still relies on this; the discovery is merely part of the initial step.
* The experiments seem quite limited, particularly with too few baseline methods for comparison (see the questions section for some suggestions).
* The assumption about invariant noise distributions across environments is rather strong, although the other assumptions seem reasonably fair.
* The overall complexity is relatively high.
* The paper lacks a conclusion section.

Please see the questions section for additional points.

**Questions:**

Overall, the paper is well written and easy to follow. The examples are very helpful. Here are some comments and questions:


* While the overview of related work in the introduction is excellent, the actual related work section seems a bit thin. Consider adding literature such as “Conditional Distributional Treatment Effect with Kernel Conditional Mean Embeddings and U-Statistic Regression” by Park et al., and “Why did the distribution change?” by Budhathoki et al. Given that you build on previous work for inferring the graph structure, you could also frame this problem as comparing it node-wise, as suggested in these papers.
* Line 112 seems to suggest that you do not support hard interventions. However, based on Eq. (3), it should be fine. Perhaps, you could clarify this more explicitly.
* Assumption A seems to be a mix of the ‘independence of mechanism’ assumption and the ‘modularity/autonomy’ assumption. The former states that the input distribution has no information about the causal mechanism, i.e., the mechanism is independent of its input (see e.g. “Information-geometric approach to inferring causal directions” by Janzing et al. for a more intuitive explanation). The ‘modularity/autonomy’ assumption states that intervening on one mechanism does not change other mechanisms in the system. I assume you refer to the latter one.
* In Remark 1, note that \tilde PA_j = PA_j from related literature could also include a node that became irrelevant post-interventions. As you mentioned below, an intervention can change the functional form. For instance, in a linear model, the coefficient can become 0 of a parent, i.e., the parent can remain in the set, it is just not a ‘minimal’ set anymore. Maybe you can emphasize that you aim for ‘causal minimality’ and then having only a subset of parents after the intervention would make sense.
* Assumption B is a quite strong, especially seeing that two environments might only change because of changes in external factors (the noise changes or becomes anomalous). Here, you would exclude these cases. In that sense, you restrict the shifts in a causal mechanism to changes in the functional form, which is significantly less general. Would be helpful to add a brief discussion about this and also emphasize the type of shift you are interested in more clearly.
* You introduce the probability mass w (line 164), but wouldn't we know with a probability of 1 from which environment an observation comes from?
* Stating that you do not rely on the knowledge about the causal structure seems slightly misleading as you still utilize Rolland et al.'s method. Hence, one could also use other methods that depend on knowing the DAG after applying the same algorithm. Maybe add a brief discussion regarding this.
* Theorem 1 is a good result. Adding some intuitive explanation as to why the variance would be greater than 0 if the node shifted would be helpful.
* The variable selection step could be more explicitly referred to as an edge pruning problem.
* For the variable selection, would one use all data points from all environments?
* Algorithm 2 seems slightly out of place. It should suffice to refer to that paper.
* The experiment section is quite brief, especially the comparison with related methods. A possible approach could be to use Rolland et al.'s method to infer the graph and then apply methods as in the previously mentioned works by Park et al. and Budhathoki et al.

--Update after rebuttal-- I have read the rebuttal and further discussed with the authors. Most of my concerns have been addressed and I raised the score by 1.

**Limitations:**

No concerns about societal impact were raised. The authors have reasonably discussed the technical limitations.

---

> ### Author Rebuttal · Authors · 2023-08-10
>
> Thanks for your valuable feedback! We next respond to your questions, and please do not hesitate to ask follow-up questions.
>
> ### Response to comments from Weaknesses section
>
> Next we only respond to the two points that are not repeated in the Questions section.
>
> > "The overall complexity is relatively high."
>
> We would like to remind two points:
>
> 1. The computational complexity depends on the choice of score estimator. In this work, we used a kernel-based estimator and, as a result, the time complexity of Algorithm 1 is in the same order as that of Rolland et al. (2022). Here, any progress on score-estimation techniques will immediately improve our algorithm.
>
> 2. Our approach does not make any assumption on the structural properties of the individual DAGs. Here, note that algorithms that aim to estimate only the structure of each individual DAG can have high computational costs without structural assumptions (e.g., sparsity, or small Markov boundary). For example, the PC algorithm runs in exponential time for dense graphs (Kalisch and Bühlman, 2007); for linear SEMs, Ghoshal & Honorio, (2018) developed an efficient peeling algorithm that runs on $\mathcal{O}(d^5)$ for dense graphs; in recent gradient-based methods, the iteration complexity is $\mathcal{O}(d^3)$ (Zheng et al., 2018) and the total time complexity is unknown. All these methods become prohibited for a large number of nodes.
>
>
> > "The paper lacks a conclusion section."
>
> Thank you for bringing this up. We will add a conclusion section for the camera ready.
>
> ### Response to comments from Questions section
>
> > "While the overview of related work ..."
>
> Thank you for suggesting these references. We will add and briefly discuss them in the revision. Regarding empirical comparisons against these approaches, we shall note that the work of Budhathoki et al. does not explicitly state which method they use for estimating the conditional distributions nor how they test for differences, moreover, they do not provide a source code. Similarly, there is no code available for the work of Park et al. and it is unclear how to reproduce them. We will be happy to provide numerical comparisons if more pointers are provided.
>
>
> > "Line 112 seems to suggest that..."
>
> Thank you for noting this! To be precise, our results also hold for stochastic hard interventions. That is, an intervention can completely remove the parent set of a node (i.e. remove all incoming edges), but we still need to consider noise to have a well-defined ANM. We will add this explicitly.
>
>
> > "Assumption A seems to be a mix of..."
>
> Great point! To be crystal clear, we assume that an environment admits a factorization of the form of eq.(3). As you correctly point out, we only need to consider modularity. We will fix this in the revision.
>
> > "In Remark 1, note that..."
>
> Note that Assumption C, which states that each functional $f_j^h(\mathrm{PA}_j^h)$ is nonlinear in every component, implicitly ensures that no irrelevant parents are included in the set $\mathrm{PA}_j^h$. Nonetheless, we will make a short footnote to discuss this for further clarity.
>
> > "Assumption B is a quite strong, ..."
>
> See our response to all reviewers.
>
> > "You introduce the probability mass w..."
>
> Correct, we know the environment membership of each sample. However, the introduction of $w_h$ is primarily to illustrate that the pooled data comes from a mixture distribution. As seen from Theorem 1, our identifiability result does not depend on $w_h$ and is helpful only for proving the theorem.
>
> > "Stating that you do not rely on the knowledge about the causal structure..."
>
> Throughout the text we carefully used the term “entire DAG structure” or “full structure” to explicitly indicate that our method does not require estimation of directed edges in the individual DAGs, which we argue is not misleading. As stated in our paper, we only use Rolland et al.’s leaf-identifiability result (Proposition 1 in our paper, or Lemma 1, item (i), in Rolland et al.), which has two important consequences:
>
> 1. Leaf identifiability only requires estimation of the *diagonal of the score’s Jacobian*, whereas estimation of the edges in the individual DAGs would require estimation of the whole score’s Jacobian matrix. Thus, our approach is much more efficient.
>
> 2. Even when a topological order is known, in finite samples, *accurate* estimation of all directed edges is very challenging for nonparametric models. For instance, in Table 3 of Rolland et al.’s paper, we can observe that SCORE, CAM and GraN-DAG, **all output a DAG with SHD of at least 130!** This means 130 edges are either missing, incorrectly oriented, or should not exist. Moreover, this high SHD value was obtained for Erdos-Renyi graphs with 50 nodes and 200 edges in expectation (an edge density of about 16%, i.e., *the graph is not dense at all*). One of the strengths of our algorithm is precisely that we do not need to estimate the edges on the individual DAGs in order to identify shifted nodes.
>
> > "Theorem 1 is a good result..."
>
> Thank you for this suggestion! We will make sure to add a short high-level explanation of the theorem for the camera ready.
>
>
> > "For the variable selection, would one use all data points from all environments?"
>
> For estimating structural changes across environments, we use data from each environment separately to estimate the parents of the *shifted nodes* and, thus, observe changes in the parent sets of the shifted nodes. We should note that estimating structural changes of shifted nodes is an optional step, it is possible that a scientist/practitioner is interested only in detecting shifted nodes, i.e., variables that have undergone a mechanism shift .
>
>
> > "Algorithm 2 seems slightly out of place..."
>
> Fair point. For completeness, we will keep Algorithm 2 and move it to the appendix
>
>
> > "The experiment section is quite brief..."
>
> We have run additional experiments including other baselines. Please see our rebuttal to all reviewers.

---

> > ### Comment · Reviewer_m244 · 2023-08-11
> >
> > I'd like to thank the authors for their detailed and thoughtful response to the questions and concerns I raised.
> >
> > > "While the overview of related work ..."
> >
> > Thank you for adding a comparison with these works. For the experiments, I'm aware of an implementation of the approach by Park et al. at: https://github.com/py-why/dodiscover/blob/main/dodiscover/cd/kernel_test.py#L13. As for the work by Budhathoki et al., based on the paper, it seems that conditional distributions were estimated using additive noise models. However, the test for mechanism change is based on testing whether X and Z are independent given Y, with X originating from one environment, Y from another, and Z being a binary indicator variable for the environment. Seeing this, any independence test could be utilized, such as a kernel independence test.
> >
> > > "Stating that you do not rely on the knowledge about the causal structure..."
> >
> > Thanks, the clarification regarding the diagonal for the leaf identification was really helpful.

---

> > > ### Author Response · Authors · 2023-08-15
> > > **Thanks for the pointers! Park et al. and Budhathoki et al. methods are added as baselines**
> > >
> > > We are glad to see our responses were helpful! Also, thank you for pointing additional resources to compare against Park et al. and Budhathoki et al., we will add both baselines to our experiments section. For brevity, see below a table containing all baselines for ER4 graphs. As can be observed, our method consistently outperforms the other baselines by a large margin.
> > >
> > > _Short remark on these two baselines:_ As per your suggestion, for the KCD method, we first use Rolland et al.'s algorithm to estimate the topological order, and then employ KCD to analyze the conditional differences; for Budhathoki's method, we initiate the process by using Rolland et al.'s method to estimate the _entire_ DAG, and then apply a kernel conditional independence test to detect any shifts in the mechanisms.
> > >
> > > Given that your main concerns on Assumption B and additional baselines are being addressed, we hope these updates to our paper will make you feel more positive about our work. If there are any further questions, please let us know!
> > >
> > >
> > > |    d |     Method |             F1 score |  Precision |     Recall  |           Time (seconds) |
> > > | ---: | ---------: | :-------------: | ---------: | ---------: | -------------: |
> > > |   10 |       Ours | **0.87 ± 0.03** |  0.81 ± 0.04 | 0.97 ± 0.02 |     31.88 ± 0.03 |
> > > |   10 | Budhathoki |     0.48 ± 0.02 | 0.32 ± 0.01 |    1.00 ± 0.00 |    33.36 ± 3.15 |
> > > |   10 |       CITE |     0.57 ± 0.02 |   0.40 ± 0.02 |    1.00 ± 0.00 |     0.39 ± 0.10 |
> > > |   10 |        DCI |      0.6 ± 0.05 | 0.52 ± 0.05 | 0.73 ± 0.07 |  0.11 ± 0.02 |
> > > |   10 |        KCD |     0.71 ± 0.06 | 0.87 ± 0.07 | 0.65 ± 0.08 |    135.2 ± 2.15 |
> > > |   10 |      SCORE |     0.48 ± 0.02 | 0.32 ± 0.01 |    1.00 ± 0.00 |     19.6 ± 7.04 |
> > > |   10 |    UT-IGSP |      0.5 ± 0.02 | 0.34 ± 0.02 |    1.00 ± 0.00 |     0.07 ± 0.01 |
> > > --
> > > |   20 |       Ours | **0.91 ± 0.02** | 0.85 ± 0.03 | 0.98 ± 0.01 |    73.16 ± 0.11 |
> > > |   20 | Budhathoki |      0.49 ± 0.02 | 0.33 ± 0.018 |    1.00 ± 0.00 |    88.56 ± 5.21 |
> > > |   20 |       CITE |     0.58 ± 0.03 | 0.41 ± 0.025 |    1.00 ± 0.00 |     1.43 ± 0.49 |
> > > |   20 |        DCI |     0.55 ± 0.03 |  0.50 ± 0.032 | 0.65 ± 0.04 |      3.27 ± 0.38 |
> > > |   20 |        KCD |     0.52 ± 0.11 | 0.68 ± 0.132 | 0.45 ± 0.10 |   310.39 ± 4.86 |
> > > |   20 |      SCORE |     0.51 ± 0.02 | 0.34 ± 0.016 |    1.00 ± 0.00 |    39.81 ± 0.30 |
> > > |   20 |    UT-IGSP |     0.52 ± 0.03 | 0.36 ± 0.024 |    1.00 ± 0.00 | 0.12 ± 0.01 |
> > > --
> > > |   30 |       Ours | **0.91 ± 0.01** |  0.85 ± 0.02 | 0.99 ± 0.01 |   124.49 ± 0.14 |
> > > |   30 | Budhathoki |     0.51 ± 0.01 | 0.34 ± 0.01 |    1.00 ± 0.00 |  190.86 ± 11.22 |
> > > |   30 |       CITE |      0.62 ± 0.02 | 0.45 ± 0.02 |    1.00 ± 0.00 |     4.81 ± 2.10 |
> > > |   30 |        DCI |     0.57 ± 0.03 | 0.49 ± 0.03 |  0.7 ± 0.03 |    27.08 ± 1.56 |
> > > |   30 |        KCD |     0.47 ± 0.07 |  0.8 ± 0.11 | 0.35 ± 0.06 |   428.97 ± 5.51 |
> > > |   30 |      SCORE |     0.52 ± 0.01 | 0.35 ± 0.01 |    1.00 ± 0.00 |  160.29 ± 16.17 |
> > > |   30 |    UT-IGSP |     0.53 ± 0.02 | 0.36 ± 0.02 |    1.00 ± 0.00 | 0.18 ± 0.01 |
> > > --
> > > |   40 |       Ours | **0.93 ± 0.01** |  0.87 ± 0.02 |    1.00 ± 0.00 |   184.61 ± 0.37 |
> > > |   40 | Budhathoki |     0.48 ± 0.02 | 0.32 ± 0.01 |    1.00 ± 0.00 |  410.82 ± 13.25 |
> > > |   40 |       CITE |     0.58 ± 0.03 | 0.41 ± 0.03 |    1.00 ± 0.00 |     6.29 ± 1.67 |
> > > |   40 |        DCI |     0.55 ± 0.03 | 0.47 ± 0.03 | 0.68 ± 0.03 |    96.77 ± 7.47 |
> > > |   40 |        KCD |     0.48 ± 0.05 | 0.78 ± 0.07 | 0.36 ± 0.05 |   586.04 ± 7.36 |
> > > |   40 |      SCORE |     0.51 ± 0.02 | 0.34 ± 0.02 |    1.00 ± 0.00 |  349.48 ± 17.04 |
> > > |   40 |    UT-IGSP |     0.53 ± 0.03 | 0.36 ± 0.02 |    1.00 ± 0.00 | 0.29 ± 0.02 |
> > > --
> > > |   50 |       Ours | **0.94 ± 0.01** | 0.88 ± 0.02 |    1.00 ± 0.00 |    249.4 ± 0.34 |
> > > |   50 | Budhathoki |      0.48 ± 0.01 | 0.31 ± 0.01 |    1.00 ± 0.00 |  765.46 ± 12.11 |
> > > |   50 |       CITE |     0.56 ± 0.02 | 0.39 ± 0.02 |    1.00 ± 0.00 |      7.99 ± 1.38 |
> > > |   50 |        DCI |      0.54 ± 0.02 | 0.48 ± 0.02 | 0.64 ± 0.02 |  244.32 ± 19.03 |
> > > |   50 |        KCD |     0.35 ± 0.05 | 0.66 ± 0.07 | 0.25 ± 0.04 |   823.88 ± 5.74 |
> > > |   50 |      SCORE |     0.48 ± 0.01 |  0.32 ± 0.01 |    1.00 ± 0.00 |  650.16 ± 26.02 |
> > > |   50 |    UT-IGSP |     0.55 ± 0.02 | 0.38 ± 0.02 |    1.00 ± 0.00 | 0.39 ± 0.03 |

---

> > > > ### Comment · Reviewer_m244 · 2023-08-15
> > > >
> > > > Thanks for adding these methods to the comparison. Although there are several points in the paper that require revision, the authors have promised to address these issues and I am willing to increase my score from 6 to 7.

---

> > > > > ### Author Response · Authors · 2023-08-20
> > > > >
> > > > > Dear reviewer, we are glad that the additional comparisons were helpful! We are really thankful for your support of our work. Rest assured, the revision will contain the promised updates.

---

### Author Rebuttal · Authors · 2023-08-10

We thank all reviewers for their time and effort put into our work. We truly appreciate your feedback and we are happy to see that your comments will lead to a stronger version of our manuscript. Overall, there is a positive feeling about the significance of the problem, our contributions, and presentation of our work. We detected two major concerns from most of the reviewers that apparently lead them to assign a borderline score: One concern is on Assumption B, and the other on the lack of additional baselines in the experiments. We next tackle both, and additionally highlight important aspects of the problem setting and our contribution that may have been overlooked. In light of the rebuttal process, we hope our answers will successfully address your concerns, and we welcome any follow-up or additional questions you may have.

### 1. On Assumption B.

TL;DR: we can relax Assumption B and allow for changes in the noise distributions, without many changes in the proof. The definition of shifted node will be updated to changes in the mechanisms (conditional distribution), and not just *functional changes*, as the mechanism shift can now stem from changes in the functional or in the noise distribution.

Nevertheless, we would like to point out that Assumption B was not unique to our work and was actually considered in the two most closely related works to ours, namely, by Wang et al. and by Ghoshal et al. The consistency result in Wang et al.’s Thm 4.4 requires invariant noises in order to orient all edges in the difference DAG. Likewise, Ghoshal et al. also assume invariant noises in their Thm 1. Motivated by these works, we almost by default assumed invariant noises and focused on identifying functional shifts.
Thanks to your criticisms on this assumption, we were challenged to think about ways to relax it, and ended up proving that without Assumption B, our method can more generally identify distribution shifts (not only functional shifts). We briefly mention the updates to be made.

Updates: Without Assumption B, Thm 1 (line 187) can be modified as follows:

* if $j$ is a leaf in all DAGs, then the conditional distribution $p^h(X_j^h \mid PA_j^h)$ is shifted if and only if $\mathrm{Var}_X[\frac{\partial s_j(x)}{\partial x_j}] > 0$.

The proof updates start at eq.(8) (L589), where the last two summation terms can only be canceled out if and only if:
 $ \frac{\partial}{\partial x_j} \log p^h_{N_j}(x_j-f_j^h(PA_j^h))=\frac{\partial}{\partial x_j}\log p^{h'}_{N_j}(x_j-f_j^{h'}(PA_j^{h'}))$ for all $h,h'\in [H].$

For brevity, consider Gaussian noises, then the above equation boils down to:
$ \frac{x_j-f_j^h(PA_j^h)}{(\sigma_j^h)^2}=\frac{x_j-f_j^{h'}(PA_j^{h'})}{(\sigma_j^{h'})^2}$, for all $x_j, PA_j^h, PA_j^{h'}.$
With some algebra, we can prove that this equation holds if and only if $\sigma_j^h=\sigma_j^{h'}, PA_j^h = PA_j^{h'}$, and $f_j^h(PA_j^h) = f_j^{h'}(PA_j^{h'}).$
Since the conditional distribution is $p^h(X_j\mid PA_j^h)\sim N(f_j^h(PA_j^h),(\sigma_j^h)^2)$, if both the mean and variance parts are the same, there are no conditional distribution shifts, and the last two summation terms in eq.(8) will be canceled out, resulting in zero variance. Otherwise, any difference in functional relationships or noise variance across environments will prevent the cancellation of the last two summation terms in eq.(8), inducing positive variance. Lastly, Lemma 2 will also be updated.

### 2. Comparison to other baselines

We include plots of our new experiments in the attached PDF. Due to space constraints, we only show F1 scores and Runtimes, but will provide all details in the revision.

**New baselines:** As per your suggestions, we ran more experiments, similar to the setting described in App. D.2, by adding three more baselines: SCORE (Rolland et al.), UT-IGSP (Permutation-Based Causal Structure Learning with Unknown Intervention Targets), and CITE (Scalable Intervention Target Estimation in Linear Models). Figures R.1 and R.2 correspond to results on shifted nodes and structural differences, respectively.

**Experiments using Gaussian Processes (GPs):** We also include experiments where the functionals are drawn from GPs. These results are in Figure R.3. We will include all experimental details in the revision, but will be happy to clarify any questions in the discussion period.

As seen in the figures, our method consistently outperforms the other baselines in terms of the F1 score, further affirming the robustness and efficiency of our approach.

### 3. Our problem setting is not the same as identifying intervention targets

While some reviewers have suggested that methods for estimating intervention targets could be applied to our problem, we would like to clarify that there is a fundamental difference between "estimating mechanism shifts" and "estimating intervention targets." Briefly, **all shifted nodes are intervened nodes, but not all intervened nodes are shifted nodes.** This is because we are interested in *changes across environments*. For example, it is possible that a variable has the same intervention across the environments, and thus, it would not be considered a shifted node. Finally, we do not assume access to observational data, which is a typical assumption for methods designed to detect intervention targets.

### 4. Important assumptions that we do not make

With the exception of Reviewer YGJy, we note that two important aspects of our approach have been largely overlooked, and we would like to reiterate their importance: (1) We do not make parametric assumptions on the functional mechanisms; and (2) We do not make any structural assumptions on the individual DAGs, such as sparsity or small Markov boundaries. Without these assumptions, the naive approach of learning the individual DAGs, fitting functionals, and then testing for changes, is doomed to fail. Not only this naive approach becomes computationally intractable, but also is statistically inefficient.

---

### Decision · Program_Chairs · 2023-09-21

**Decision:**

Accept (poster)

**Comment:**

The paper provides a clear to read framework, makes validated algorithmic contributions and is a pleasure to read. All reviewers are in favor of acceptance. While there were considerable (positive) discussions with the reviewers and it is highly recommended and assumed that the authors include the take-aways and improvements as well as the additional results into the final version of the paper.